# RIZE: Adaptive Regularization for Imitation Learning

**Adib Karimi**[†]                                                    *adibkarimi23@aut.ac.ir*
*Department of Computer Engineering*
*Amirkabir University of Technology*

**Mohammad Mehdi Ebadzadeh**                                         *ebadzadeh@aut.ac.ir*
*Department of Computer Engineering*
*Amirkabir University of Technology*

**Reviewed on OpenReview:** *https://openreview.net/forum?id=a6DWqXJZCZ*

## Abstract

We propose a novel Inverse Reinforcement Learning (IRL) method that mitigates the rigidity of fixed reward structures and the limited flexibility of implicit reward regularization. Building on the Maximum Entropy IRL framework, our approach incorporates a squared temporal-difference (TD) regularizer with adaptive targets that evolve dynamically during training, thereby imposing adaptive bounds on recovered rewards and promoting robust decision-making. To capture richer return information, we integrate distributional RL into the learning process. Empirically, our method achieves expert-level performance on complex MuJoCo and Adroit environments, surpassing baseline methods on the `Humanoid-v2` task with limited expert demonstrations. Extensive experiments and ablation studies further validate the effectiveness of the approach and provide insights into reward dynamics in imitation learning. Our source code is available at `https://github.com/adibka/RIZE`.

## 1 Introduction

Designing effective reward functions remains a fundamental challenge in Reinforcement Learning (RL). While dense, well-shaped rewards can facilitate learning, they often require laborious task-specific tuning and domain expertise, limiting their scalability (Amodei et al., 2016; Peng et al., 2020). To circumvent these limitations, researchers have explored alternative approaches, including sparse rewards upon task completion (Silver et al., 2016), learning from expert trajectories (Schaal, 1996; Ng & Russell, 2000; Osa et al., 2018), human preference-based reward modeling (Christiano et al., 2017; Lee et al., 2021; Hejna & Sadigh, 2023), and intrinsically motivated RL (Oudeyer et al., 2007; Schmidhuber, 2010; Colas et al., 2022). Among these, Inverse Reinforcement Learning (IRL) (Abbeel & Ng, 2004) offers a compelling alternative by inferring reward functions directly from expert demonstrations, bypassing manual reward engineering. IRL has driven breakthroughs in robotics (Osa et al., 2018), autonomous driving (Knox et al., 2023), and drug discovery (Ai et al., 2024).

A prominent framework in IRL is Maximum Entropy (MaxEnt) IRL (Ziebart, 2010), which underpins many state-of-the-art (SOTA) IL methods. Prior works have combined MaxEnt IRL with adversarial training (Ho & Ermon, 2016; Fu et al., 2018) to minimize divergences between agent and expert distributions. However, these adversarial methods often suffer from instability during training. To address this, recent research has introduced implicit reward regularization, which indirectly represents rewards via Q-values by inverting the Bellman equation. For instance, IQ-Learn (Garg et al., 2021) unifies reward and policy representations using Q-functions with an $L_2$-norm regularization on rewards, while LSIQ (Al-Hafez et al., 2023) minimizes the chi-squared divergence between expert and mixture distributions, resulting in a squared temporal difference (TD) error objective analogous to SQIL (Reddy et al., 2020). Despite its effectiveness, this method has limitations:

---

[†]Corresponding author.

LSIQ assigns fixed targets for implicit rewards (e.g., +1 for expert samples and -1 for agent samples), which constrains flexibility by treating all tasks and state-action pairs uniformly, limiting performance and requiring additional gradient steps for convergence.

We propose an extension of implicit reward regularization under the MaxEnt IRL framework, introducing two key advancements. **Adaptive Targets:** We enhance prior TD-error regularization by introducing learnable targets $\lambda^{\pi_E}$ and $\lambda^{\pi}$ that dynamically adjust during training. These targets replace static constraints with context-sensitive reward alignment, preventing rewards from over-increasing or over-decreasing. Crucially, our theoretical analysis reveals that these adaptive bounds constrain implicit rewards to well-defined ranges $\left[ -\frac{1}{2c} + \min\left\{ \lambda^{\pi_E}, \lambda^{\pi} \right\}, \ \frac{1}{2c} + \max\left\{ \lambda^{\pi_E}, \lambda^{\pi} \right\} \right]$, where $c$ is the regularization coefficient. Since implicit rewards derive from Q-values, this regularization indirectly stabilizes policy training. **Distributional RL Integration:** We incorporate return distributions $Z^{\pi}(s, a)$ (Bellemare et al., 2017) to capture richer uncertainty information in returns, while using their expectations for policy optimization. Though distributional RL has shown success in adversarial IRL (Zhou et al., 2023), it remains unexplored in non-adversarial settings. Our work bridges this gap, demonstrating its efficacy in MaxEnt IRL while preserving theoretical guarantees. Unifying these advances, our framework outperforms IL baselines on MuJoCo (Todorov et al., 2012) and Adroit (Rajeswaran et al., 2018) benchmarks. Notably, our approach shows clear benefits on complex tasks like `Humanoid-v2` and `Hammer-v1`, where adaptive targets and distributional learning improve training stability, as confirmed by our ablations.

Our contributions are threefold. First, we introduce adaptive targets for implicit reward regularization, enabling dynamic reward bounds that enhance stability and alignment during training. Second, we integrate return distributions into implicit reward frameworks, capturing richer uncertainty information while preserving theoretical consistency. Third, we empirically validate our approach through extensive experiments on MuJoCo and Adroit tasks, demonstrating superior performance in complex environments.

## 2 Related Work

Imitation learning (IL) and inverse reinforcement learning (IRL) (Watson et al., 2023) are foundational paradigms for training agents to mimic expert behavior from demonstrations. Behavioral Cloning (BC) (Pomerleau, 1991), the simplest IL approach, treats imitation as a supervised learning problem by directly mapping states to expert actions. While computationally efficient, BC is prone to compounding errors (Ross & Bagnell, 2011) due to covariate shift during deployment. The Maximum Entropy IRL framework (Ziebart, 2010) addresses this limitation by probabilistically modeling expert behavior as reward maximization under an entropy regularization constraint, establishing a theoretical foundation for modern IRL methods. The advent of adversarial training marked a pivotal shift in IL methodologies. Ho & Ermon (2016) introduced Generative Adversarial Imitation Learning (GAIL), which formulates imitation learning as a generative adversarial game (Goodfellow et al., 2014) where an agent learns a policy indistinguishable from the expert's by minimizing the Jensen–Shannon divergence between their state–action distributions. This framework was generalized by Ghasemipour et al. (2019) in f-GAIL, which replaces the Jensen–Shannon divergence with arbitrary (f)-divergences to broaden applicability. Concurrently, Kostrikov et al. (2019) proposed Discriminator Actor–Critic (DAC), improving sample efficiency via off-policy updates and terminal-state reward modeling while mitigating reward bias. Most recently, Chang et al. (2024) cast adversarial imitation as a boosting procedure: AILBoost maintains an ensemble of weighted policies and trains the discriminator against a weighted replay buffer approximating the ensemble's occupancy, enabling fully off-policy training and reporting gains over DAC on DeepMind Control tasks (Tassa et al., 2018).

Recent advances have shifted toward methods that bypass explicit reward function estimation. Kostrikov et al. (2020) introduced ValueDICE, an offline IL method that leverages an inverse Bellman operator to avoid adversarial optimization. Similarly, Garg et al. (2021) developed IQ-Learn, which circumvents the challenges of MaxEnt IRL by optimizing implicit rewards derived directly from expert Q-values. A parallel research direction simplifies reward engineering by assigning fixed rewards to expert and agent samples. Reddy et al. (2020) pioneered this approach with Soft Q Imitation Learning (SQIL), which assigns binary rewards to transitions from expert and agent trajectories. Most recently, Al-Hafez et al. (2023) proposed Least Squares Inverse Q-Learning (LSIQ), enhancing regularization by minimizing chi-squared divergence between expert

and mixture distributions while explicitly managing absorbing states through critic regularization. In the same spirit of avoiding adversarial training, Coherent Soft Imitation Learning (CSIL) (Watson et al., 2023) inverts the soft policy update to derive an explicit shaped "coherent" reward from a behavior-cloned policy and then fine-tunes the policy with standard RL using online or offline data. Orthogonally, Jain et al. (2025) introduce Successor Feature Matching, a non-adversarial IRL method that forgoes explicit reward learning by directly matching expert and learner successor features, and notably supports state-only demonstrations. Complementing these, Wu et al. (2025) propose a diffusion-based framework that learns score functions on expert and learner states and optimizes a score-difference cost (a diffusion score divergence), offering a non-adversarial alternative that also works with state-only demos. Our work builds on IQ-Learn (Garg et al., 2021) and LSIQ (Al-Hafez et al., 2023) by applying a squared TD regularizer with adaptive targets and by employing an Implicit Quantile Network (IQN) critic (Dabney et al., 2018a).

## 3 Background

### 3.1 Preliminary

We consider a Markov Decision Process (MDP) (Puterman, 2014) to model policy learning in Reinforcement Learning (RL). The MDP framework is defined by the tuple $\langle \mathcal{S}, \mathcal{A}, p_0, P, R, \gamma \rangle$, where $\mathcal{S}$ denotes the state space, $\mathcal{A}$ the action space, $p_0$ the initial state distribution, $P : \mathcal{S} \times \mathcal{A} \times \mathcal{S} \to [0, 1]$ the transition kernel with $P(\cdot \mid s, a)$ specifying the likelihood of transitioning from state $s$ given action $a$, $R : \mathcal{S} \times \mathcal{A} \to \mathbb{R}$ the reward function, and $\gamma \in [0, 1]$ the discount factor which tempers future rewards. A stationary policy $\pi \in \Pi$ is characterized as a mapping from states $s \in \mathcal{S}$ to distributions over actions $a \in \mathcal{A}$. The primary objective in RL (Sutton & Barto, 2018) is to maximize the expected sum of discounted rewards, expressed as $\mathbb{E}_\pi \left[ \sum_{t=0}^{\infty} \gamma^t R(s_t, a_t) \right]$. Furthermore, the occupancy measure $\rho_\pi(s, a)$ for a policy $\pi \in \Pi$ is given by $(1 - \gamma)\pi(a \mid s) \sum_{t=0}^{\infty} \gamma^t P(s_t = s \mid \pi)$. The corresponding measure for an expert policy, $\pi_E$, is similarly denoted by $\rho_E$. In Imitation Learning (IL), the expert policy $\pi_E$ is typically unknown, and only a finite set of expert demonstrations is available, rather than explicit reward feedback from the environment.

### 3.2 Distributional Reinforcement Learning

Maximum Entropy (MaxEnt) RL (Haarnoja et al., 2018) focuses on addressing the stochastic nature of action selection by maximizing the entropy of the policy, while Distributional RL (Bellemare et al., 2017) emphasizes capturing the inherent randomness in returns. Combining these perspectives, the distributional soft value function $Z : \mathcal{S} \times \mathcal{A} \to \mathcal{Z}$ (Ma et al., 2020) for a policy $\pi \in \Pi$ encapsulates uncertainty in both rewards and actions, with $\mathcal{Z}$ representing the space of return distributions. It is formally defined as:

$$Z(s, a) = \sum_{t=0}^{\infty} \gamma^t [R(s_t, a_t) + \alpha \mathcal{H}(\pi(\cdot \mid s_t))], \tag{1}$$

where $\mathcal{H}(\pi) = \mathbb{E}_\pi[-\log \pi(a \mid s)]$ denotes the entropy of the policy, and $\alpha > 0$ balances entropy with reward.

The distributional soft Bellman operator $\mathcal{B}_D^\pi : \mathcal{Z} \to \mathcal{Z}$ for a given policy $\pi$ is introduced as $(\mathcal{B}_D^\pi Z)(s, a) \overset{D}{=} R(s, a) + \gamma[Z(s', a') - \alpha \log \pi(a' \mid s')]$, where $s' \sim P(\cdot \mid s, a)$, $a' \sim \pi(\cdot \mid s')$, and $\overset{D}{=}$ signifies equality in distribution. Notably, this operator exhibits contraction properties under the p-Wasserstein metric, ensuring convergence to a unique fixed point, the distributional soft return function for the given policy.

A practical approach to approximating the return distribution $Z$ involves modeling its quantile function $F_Z^{-1}(\tau)$, evaluated at specific quantile levels $\tau \in [0, 1]$ (Dabney et al., 2018b). The quantile function is defined as $F_Z^{-1}(\tau) = \inf\{z \in \mathbb{R} : \tau \leq F_Z(z)\}$, where $F_Z(z) = \mathbb{P}(Z \leq z)$ is the cumulative distribution function of $Z$. For simplicity, we denote the quantile-based representation as $Z_\tau(s, a) := F_Z^{-1}(\tau)$. To discretize this representation, we define a sequence of quantile levels, denoted as $\{\tau_i\}_{i=0,\dots,N-1}$, where $0 = \tau_0 < \dots < \tau_{N-1} = 1$. These quantiles partition the unit interval into $N$ fractions. For uniformly sampled quantiles $\tau \sim U(0, 1)$, $Z_\tau(s, a)$ denotes the $\tau$-quantile (scalar) of the return distribution.

### 3.3 Inverse Reinforcement Learning

Given expert trajectory data, Maximum Entropy (MaxEnt) Inverse RL (Ziebart, 2010) aims to infer a reward function $R(s, a)$ from the family $\mathcal{R} = \mathbb{R}^{S \times A}$. Instead of assuming a deterministic expert policy, this method optimizes for stochastic policies $\pi \in \Pi$ that maximize $R$ while matching expert behavior. GAIL (Ho & Ermon, 2016) extends this framework by introducing a convex reward regularizer $\psi : \mathbb{R}^{S \times A} \to \bar{\mathbb{R}}$, leading to the adversarial objective:

$$\max_{R \in \mathcal{R}} \min_{\pi \in \Pi} L(\pi, R) = \mathbb{E}_{\rho_E}[R(s, a)] - \mathbb{E}_{\rho_\pi}[R(s, a)] - \mathcal{H}(\pi) - \psi(R). \tag{2}$$

IQ-Learn (Garg et al., 2021) departs from adversarial training by implicitly representing rewards through Q-functions $Q \in \Omega$ (Piot et al., 2014). It leverages the inverse soft Bellman operator $\mathcal{T}^\pi$, defined as:

$$(\mathcal{T}^\pi Q)(s, a) = Q(s, a) - \gamma \mathbb{E}_{s' \sim P(\cdot|s,a), a' \sim \pi(\cdot|s')}[Q(s', a') - \alpha \log \pi(a'|s')]. \tag{3}$$

For a fixed policy $\pi$, $\mathcal{T}^\pi$ is bijective, ensuring a one-to-one correspondence between $Q$-values and rewards: $\mathcal{T}^\pi Q = R$ and $Q = (\mathcal{T}^\pi)^{-1} R$. This allows reframing the MaxEnt IRL objective (2) in Q-policy space as $\max_{Q \in \Omega} \min_{\pi \in \Pi} \mathcal{J}(\pi, Q)$. IQ-Learn simplifies the problem by defining the implicit reward $R_Q(s, a) = \mathcal{T}^\pi Q(s, a)$ and applying an L2 regularizer $\psi(R_Q)$. The final objective becomes:

$$\begin{aligned} \max_{Q \in \Omega} \min_{\pi \in \Pi} \mathcal{J}(\pi, Q) = \ & \mathbb{E}_{\rho_E}[R_Q(s, a)] - \mathbb{E}_{\rho_\pi}[R_Q(s, a)] - \alpha \mathcal{H}(\pi) \\ & - c \Big[ \mathbb{E}_{\rho_E}[R_Q(s, a)^2] + \mathbb{E}_{\rho_\pi}[R_Q(s, a)^2] \Big]. \end{aligned} \tag{4}$$

## 4 Methodology

This section introduces a framework that integrates Distributional Reinforcement Learning with Inverse RL. We first show how return distributions can replace point-estimate critics. We then propose an adaptive regularization technique for implicit rewards and analyze its properties. Finally, we derive RIZE, which combines distributional critics with bounded-reward imitation learning.

### 4.1 Distributional Value Integration

Our approach departs from traditional imitation learning by explicitly using return distributions as critics within an actor-critic framework (Zhou et al., 2023). We argue that learning the soft return distribution $Z(s, a)$ in Equation (1), rather than relying solely on point estimates like $Q(s, a)$, enhances decision-making by capturing uncertainty in complex environments. This view is consistent with recent neuroscience findings suggesting that decision-making in the prefrontal cortex relies on learning distributions over outcomes rather than only their expectations (Muller et al., 2024).

Moreover, access to the full return distribution enables the computation of statistical moments—most notably the expectation—which we optimize both policy and critic using the expectation of $Z$ (Bellemare et al., 2017; Dabney et al., 2018b;a), yielding a more robust learning signal while remaining compatible with IQ-Learn.

We compute $Q$ as the expectation of the soft return distribution:

$$Q(s, a) = \sum_{i=0}^{N-1} (\tau_{i+1} - \tau_i) Z_{\tau_i}(s, a), \tag{5}$$

where $\{\tau_i\}$ are quantile fractions and $Z_{\tau_i}(s, a)$ denotes the corresponding quantile values (see Lemma A.1).

### 4.2 Implicit Reward Regularization

In this section, we propose a regularizer for inverse RL that refines existing implicit reward formulations (Garg et al., 2021). The implicit reward is defined as:

$$R_Q(s, a) = Q(s, a) - \gamma \mathbb{E}_{P, \pi}[Q(s', a') - \alpha \log \pi(a'|s')]. \tag{6}$$

Previous works typically regularize implicit rewards either using $L_2$-norms (Garg et al., 2021) or by treating them as squared-TD errors between rewards and fixed targets (Reddy et al., 2020; Al-Hafez et al., 2023). While we adopt a similar squared-TD setting, we introduce adaptive targets $\lambda^{\pi_E}$ (for the expert $\pi_E$) and $\lambda^\pi$ (for the imitation policy $\pi$) to construct our convex regularizer $\Gamma : \mathbb{R}^{S \times A} \to \bar{\mathbb{R}}$:

$$\Gamma(R_Q, \lambda) = \mathbb{E}_{\rho_E}\left[(R_Q(s,a) - \lambda^{\pi_E})^2\right] + \mathbb{E}_{\rho_\pi}\left[(R_Q(s,a) - \lambda^\pi)^2\right]. \tag{7}$$

These targets self-update through a feedback loop where reward estimates continuously adapt to match moving targets:

$$\min_{\lambda^{\pi_E}} \mathbb{E}_{\rho_E}\left[(R_Q(s,a) - \lambda^{\pi_E})^2\right], \qquad \min_{\lambda^\pi} \mathbb{E}_{\rho_\pi}\left[(R_Q(s,a) - \lambda^\pi)^2\right]. \tag{8}$$

Substituting $\Gamma(R_Q, \lambda)$ for the $L_2$ term in Equation (4) and using Equation (5) to compute $Q$, we obtain:

$$\begin{aligned} \mathcal{L}(\pi, Q) = &\, \mathbb{E}_{\rho_E}[R_Q(s,a)] - \mathbb{E}_{\rho_\pi}[R_Q(s,a)] - \alpha\mathcal{H}(\pi) \\ &- c\Big[\mathbb{E}_{\rho_E}\left[(R_Q(s,a) - \lambda^{\pi_E})^2\right] + \mathbb{E}_{\rho_\pi}\left[(R_Q(s,a) - \lambda^\pi)^2\right]\Big], \end{aligned} \tag{9}$$

where $c$ is the regularization coefficient.

As in IQ-Learn and LSIQ, our method seeks behavior indistinguishable from expert demonstrations in a non-adversarial implicit-reward setting. Prior work analyzes optimal implicit rewards, drawing on Max–Min analyses from GANs (Al-Hafez et al., 2023; Goodfellow et al., 2014). Because we bind rewards to adaptive targets via $\Gamma(R_Q, \lambda)$ (Equation (7)), we carry out an analogous analysis, stated below.

**Proposition 4.1.** *Let $R_Q(s,a) = (\mathcal{T}^\pi Q)(s,a)$ denote the implicit reward derived from point-estimate $Q$-values, where $Q(s,a) = \mathbb{E}[Z(s,a)]$. Let $\rho_E(s,a)$ and $\rho_\pi(s,a)$ denote occupancy measures under $\pi_E$ and $\pi$, respectively. For fixed $\pi$, the optimal TD-regularized reward satisfies:*

$$R_Q^*(s,a) = \frac{\rho_E(s,a) - \rho_\pi(s,a)}{(2c)\left(\rho_E(s,a) + \rho_\pi(s,a)\right)} + \frac{\rho_E(s,a)\lambda^{\pi_E} + \rho_\pi(s,a)\lambda^\pi}{\rho_E(s,a) + \rho_\pi(s,a)}. \tag{10}$$

*Proof.* Differentiating $\mathcal{L}(\pi, Q)$ in Equation (9) with respect to $R_Q(s,a)$ (holding $\pi$ fixed) and setting the result to zero yields

$$0 = \rho_E - \rho_\pi - 2c\big[\rho_E(R_Q - \lambda^{\pi_E}) + \rho_\pi(R_Q - \lambda^\pi)\big], \tag{11}$$

from which Equation (10) follows. $\square$

**Corollary 4.2.** *The optimal implicit reward satisfies:*

$$R_Q^*(s,a) \in \left[-\frac{1}{2c} + \lambda_{\min}, \frac{1}{2c} + \lambda_{\max}\right], \tag{12}$$

*where $\lambda_{\min} := \min\{\lambda^{\pi_E}, \lambda^\pi\}$ and $\lambda_{\max} := \max\{\lambda^{\pi_E}, \lambda^\pi\}$. The coefficient $c$ and adaptive targets bound rewards within a well-defined range.*

*Proof.* Considering the optimal reward Equation (10), the first term lies in $\left[-\frac{1}{2c}, \frac{1}{2c}\right]$ since $\rho_E, \rho_\pi \geq 0$ (achieved when either $\rho_\pi \to 0$ or $\rho_E \to 0$). The second term is a convex combination of $\lambda^{\pi_E}$ and $\lambda^\pi$, thus in $[\lambda_{\min}, \lambda_{\max}]$. Combining intervals yields the result. In practice, we use $\lambda \in [5, 10]$ and $c \in [0.1, 0.5]$, which we found to promote stable training (see Appendix C.7). $\square$

Moreover, when the occupancy measures match, the optimal reward simplifies significantly, as stated in Corollary A.2:

$$R_Q^*(s,a) = \lambda^{\pi_E} = \lambda^\pi. \tag{13}$$

Recalling that we represent rewards through Q-values, the boundedness of the optimal reward ensures that critic updates remain constrained and—because the policy directly depends on these critic values—promotes stable policy optimization. However, as previously noted (Al-Hafez et al., 2023; Viano et al., 2022), the convergence guarantee originally stated for IQ-Learn (Garg et al., 2021) does *not* extend to the $\chi^2$-regularizer used in practice. Consequently, a formal proof for the resulting alternating SAC updates remains an open question and is left for future work.

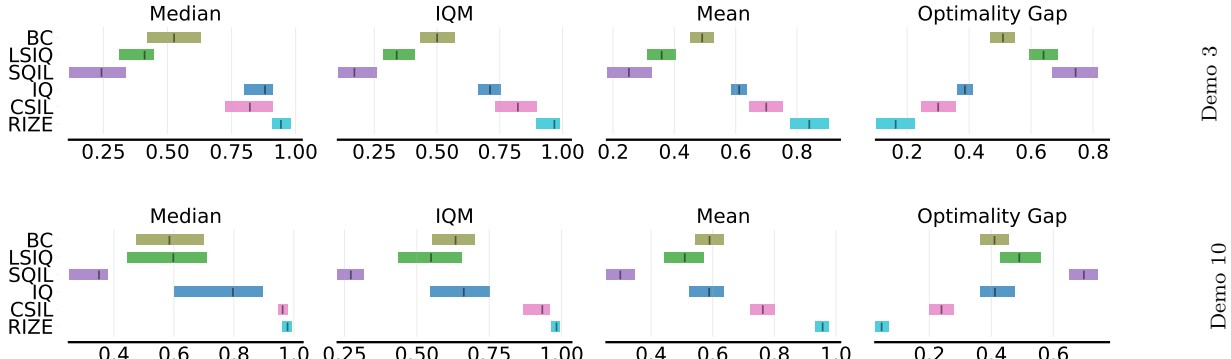

Figure 1: **RLiable** (Agarwal et al., 2021) plots for RIZE vs. BC, LSIQ, SQIL, CSIL, and IQ-Learn on six MuJoCo/Adroit tasks. For each setting (3 demos; 10 demos), we report aggregate *Median*, *IQM*, *Mean*, and *Optimality Gap* with 95% confidence intervals computed via percentile bootstrap stratified over tasks and five seeds. Scores are normalized to expert performance. Higher is better for Median, IQM, and Mean; lower is better for Optimality Gap.

### 4.3 Practical Algorithm

We now introduce **RIZE**: Adaptive Regularization for Imitation Learning (Algorithm 1). We approximate $Z$ and $\pi$ with neural networks, optimizing via $Q = \mathbb{E}[Z]$. Following Distributional SAC (Ma et al., 2020), we employ target policies for next-action sampling and a double-critic architecture with target networks for stability. Lower learning rates proved critical for robust policy updates, with a four-layer MLP policy network necessary for complex tasks.

Targets $\lambda^{\pi_E}$ and $\lambda^{\pi}$ are optimized to match expected rewards. We initialize $\lambda^{\pi_E}$ higher than $\lambda^{\pi}$ and update $\lambda^{\pi}$ with lower learning rates due to its sensitivity.

---

**Algorithm 1** RIZE

1: **Initialize** $Z_\phi$, $\pi_\theta$, $\lambda^{\pi_E}$, and $\lambda^{\pi}$
2: **for** step $t$ in $\{1, \dots, N\}$ **do**
3:     **Calculate** $Q(s,a) = \mathbb{E}[Z_\phi(s,a)]$ using Eq. 5
4:     **Update** $Z_\phi$ using Eq. 9
5:     $\phi_{t+1} \leftarrow \phi_t - \beta_Z \nabla_\phi [-\mathcal{L}(\phi)]$
6:     **Update** $\pi_\theta$ (like SAC)
7:     $\theta_{t+1} \leftarrow \theta_t + \beta_\pi \nabla_\theta \mathbb{E}_{\substack{s \sim \mathcal{D}, \\ a \sim \pi_\theta(\cdot|s)}} \big[ \min_{k=1,2} Q_k(s,a)$
                                                               $- \alpha \log \pi_\theta(a|s)]$
8:     **Update** $\lambda^{\pi}$ and $\lambda^{\pi_E}$ using Eq. 8
9:     $\lambda^{\pi}_{t+1} \leftarrow \lambda^{\pi}_t - \beta_{\lambda^{\pi}} \nabla_{\lambda^{\pi}} \Gamma(R_Q, \lambda)$
10:     $\lambda^{\pi_E}_{t+1} \leftarrow \lambda^{\pi_E}_t - \beta_{\lambda^{\pi_E}} \nabla_{\lambda^{\pi_E}} \Gamma(R_Q, \lambda)$
11: **end for**

---

## 5 Experiments

We study continuous-control imitation learning from state–action expert samples, evaluating our algorithm on five MuJoCo (Todorov et al., 2012) benchmarks (`HalfCheetah-v2, Walker2d-v2, Ant-v2, Humanoid-v2, Hopper-v2`) and one Adroit Hand task (`Hammer-v1`). We compare against state-of-the-art baselines IQ-Learn (Garg et al., 2021), LSIQ (Al-Hafez et al., 2023), SQIL (Reddy et al., 2020), CSIL (Watson et al., 2023) and Behavior Cloning (BC) (Pomerleau, 1991). All experiments use five random seeds (Henderson et al., 2018). We assess each method with three and ten expert trajectories. We report mean $\pm$ 95% confidence intervals (CI) across five seeds. Episode returns are normalized by expert performance. For implementation details, additional experimental results, and visualizations, see Appendix B and C.

**Main Results.** Figure 1 reports aggregate metrics computed with **RLiable** (Agarwal et al., 2021), which provides statistically principled evaluation via Median, Interquartile Mean (IQM), Mean, and Optimality Gap with stratified bootstrap confidence intervals; we adopt this protocol throughout, consistent with recent IL practice (e.g., AILBoost (Chang et al., 2024); SFM (Jain et al., 2025)). Across both 3- and 10-demonstration settings, RIZE achieves higher Median, IQM, and Mean and a lower Optimality Gap than the baselines; the

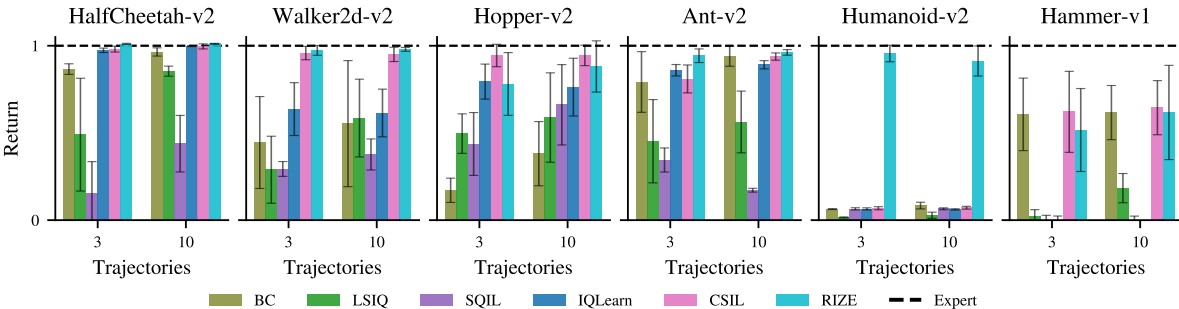

Figure 2: Normalized returns on MuJoCo and Adroit tasks for RIZE and baselines. We first compute, per seed, the average episodic return over the final third of training steps; bars show the mean across five seeds and error bars denote the 95% confidence interval. Returns are normalized to expert performance and reported for both 3 and 10 expert demonstrations.

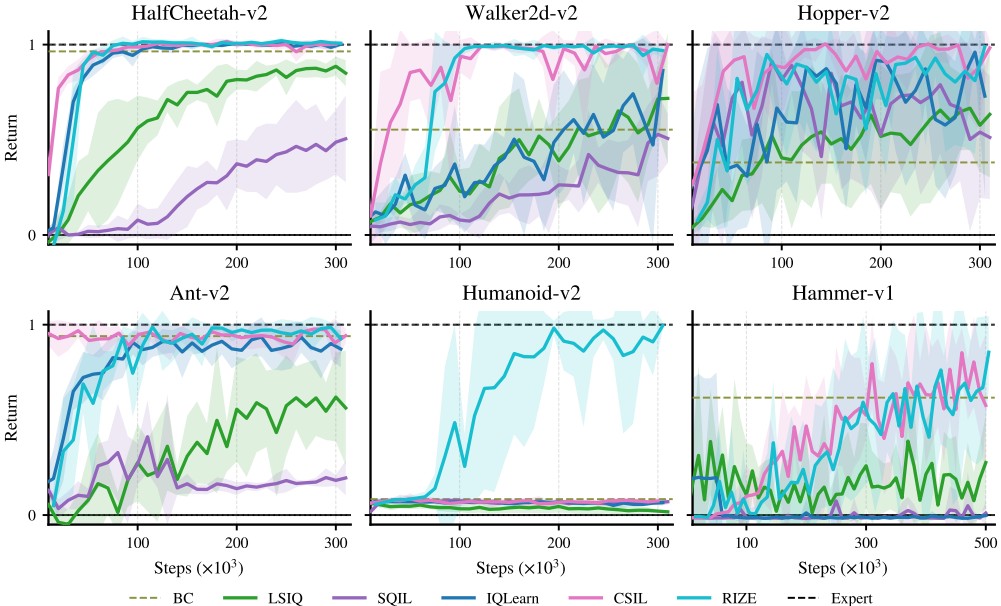

Figure 3: Learning curves on MuJoCo and Adroit tasks with 10 expert demonstrations. Lines show the mean normalized return across five seeds; shaded regions denote 95% confidence intervals.

only competitive methods overall are CSIL and IQ-Learn. Increasing the number of expert trajectories from 3 to 10 consistently improves RIZE on all RLiable aggregates. Task-wise, RIZE is the only method that solves `Humanoid-v2` (all baselines fail), while on `Hammer-v1` RIZE, CSIL, and BC show higher returns than the remaining methods. In terms of sample efficiency, CSIL often attains strong returns early—benefiting from behavior-cloning initialization—but when BC is ineffective (e.g., `Humanoid-v2`), CSIL underperforms whereas RIZE ultimately succeeds. These trends are corroborated by the bar plot in Figure 2 (average of the final third of training) and the learning curves in Figure 3. For learning curves with three demonstrations, see Figure 6.

**Recovered Rewards.** In this section, we assess whether our choice of regularizer $\Gamma(R_Q, \lambda)$ in Equation (7) with adaptive targets can effectively bound the implicit rewards. By Corollary 4.2 together with Equation (12), the optimal implicit reward is confined to the interval $\left[-\frac{1}{2c} + \lambda_{\min}, \ \frac{1}{2c} + \lambda_{\max}\right]$. Since expert rewards are supposed to be higher than those of the learner during training (based on IRL objective

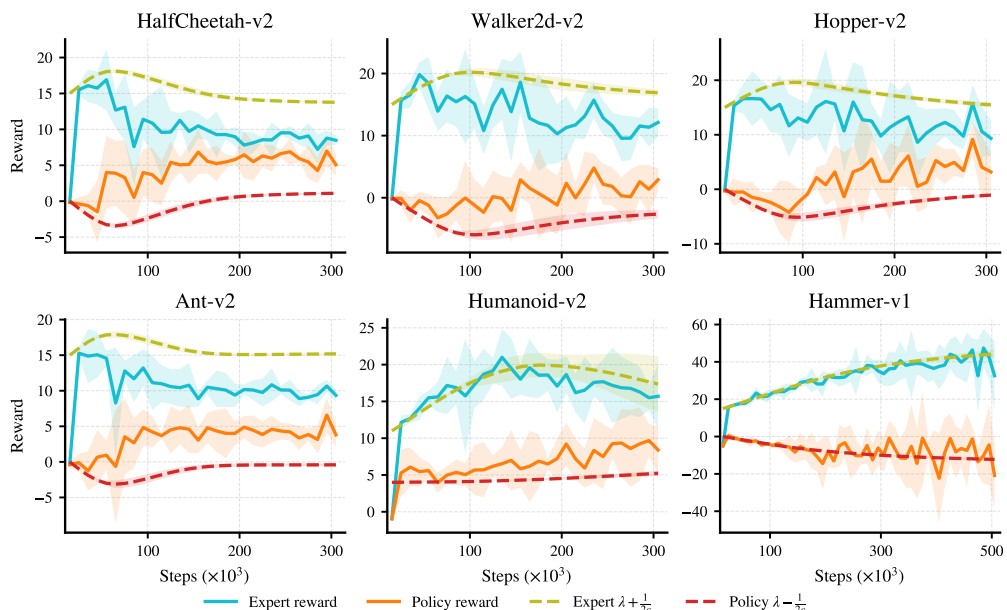

Figure 4: Implicit reward curves for expert and policy samples on MuJoCo and Adroit tasks with 10 expert demonstrations. Each subplot reports the mean across five seeds, with shaded regions showing the 95% confidence interval. Theoretical upper and lower bounds derived in this work are overlaid as separate curves in each subplot.

loss 4), and the adaptive targets track these levels, we empirically observe $\lambda^{\pi_E} \geq \lambda^{\pi}$; hence $\lambda_{\max} = \lambda^{\pi_E}$ and $\lambda_{\min} = \lambda^{\pi}$. In practice, we should see rewards confined to $\left[ -\frac{1}{2c} + \lambda^{\pi}, \frac{1}{2c} + \lambda^{\pi_E} \right]$.

Figure 4 depicts the recovered reward trajectories together with the theoretical upper and lower bounds. Across tasks, the curves remain within the predicted band, indicating that our regularizer with adaptive targets effectively bounds both expert and policy rewards. We also observe that the position and width of the band vary with the task and data regime, reflecting the flexibility of the adaptive targets to adjust to different dynamics. In contrast, *fixed* (non-adaptive) targets lack this ability, leading either to overly loose bounds or to target–reward mismatch. Overall, these results support the role of $\Gamma(R_Q, \lambda)$ and the adaptive targets in producing stable, well-calibrated implicit rewards that align with Corollary 4.2. For the 3-demonstration setting, Figure 7 shows reward trajectories with their theoretical bounds; for cross-method comparisons of reward trajectories, see Figures 9, 8, and 10a.

**Ablation on Critic Architecture.** To assess the effect of modeling the return distribution, we replace the IQN-based critic $Z(s, a)$ (Dabney et al., 2018a) in RIZE with a point-estimate $Q(s, a)$ critic and compare against the original implementation. Here, $Z(s, a)$ denotes the full distribution of discounted returns whose expectation yields $Q(s, a)$. As shown in Figure 5, the IQN critic consistently outperforms the $Q$-network across all tasks, exhibiting lower variance and greater sample efficiency. Notably, on `Humanoid-v2` and `Hammer-v1`, the $Q$-network fails to match expert performance, whereas the IQN critic maintains expert-level returns.

See Figure 13 for reward trajectories with theoretical bounds when using a $Q$-network in place of the IQN critic, and Figures 11, 12, and 10b for critic value–estimation curves on MuJoCo and Adroit tasks.

# 6  Conclusion

We propose a novel IRL framework that overcomes the limitations of fixed reward mechanisms through dynamic reward adaptation and context-sensitive regularization. Our approach ensures bounded implicit rewards and stable value function updates, leading to robust policy optimization. By integrating distributional

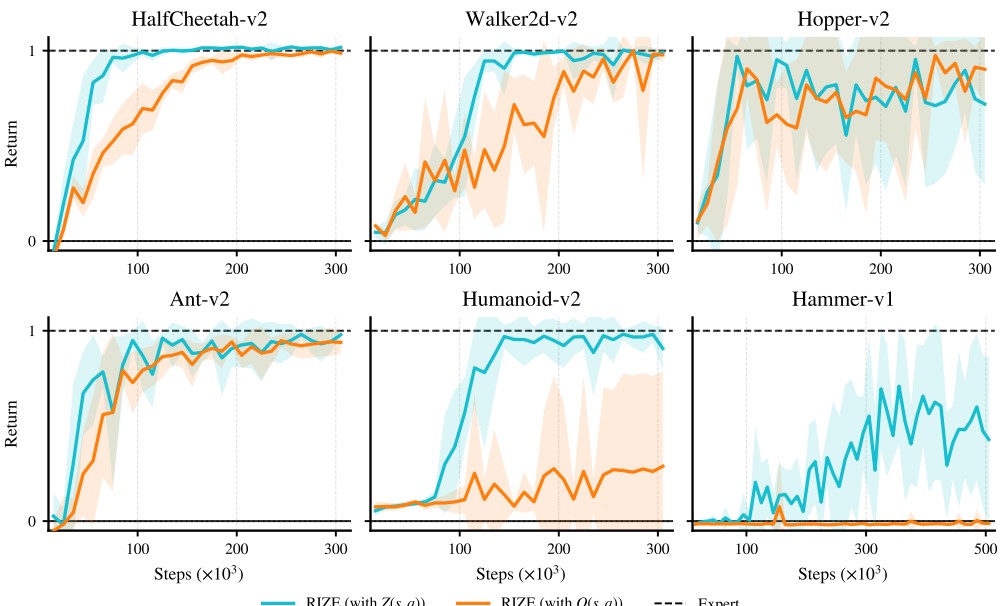

Figure 5: Ablation on critic architecture: $Z(s,a)$ via Implicit Quantile Networks (IQN) (Dabney et al., 2018a) versus classic $Q(s,a)$. We report expert–normalized returns across all MuJoCo and Adroit tasks using three expert demonstrations; metrics show the mean over five seeds with 95% confidence intervals.

RL with implicit reward learning, we capture richer return dynamics while preserving theoretical guarantees. Empirical results on MuJoCo and Adroit benchmarks show expert-like proficiency on the `Humanoid-v2` and `Hammer-v1` tasks with three expert demonstrations. Ablation studies confirm the important role of our regularization mechanism. This work unifies implicit reward regularization with distributional return representations, offering a scalable and sample-efficient solution for complex decision-making. Future directions include extending this framework to offline IL and risk-sensitive robotics, where TD regularizer with adaptive targets and uncertainty-aware return distributions can further improve robustness and generalization.

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

## A  Supporting Proofs

### A.1  Expectation of the Return Distribution

**Lemma A.1.** *The expectation of the distributional return satisfies*

$$\mathbb{E}[Z(s,a)] = Q(s,a).$$

*Proof.* From the soft return distribution definition

$$Z(s,a) = \sum_{t=0}^{\infty} \gamma^t \left[ R(s_t, a_t) + \alpha \mathcal{H}(\pi(\cdot \mid s_t)) \right], \tag{14}$$

taking expectations yields:

$$
\begin{aligned}
\mathbb{E}[Z(s,a)] &= \mathbb{E}\left[ \sum_{t=0}^{\infty} \gamma^t \left( R(s_t, a_t) + \alpha \mathcal{H}(\pi(\cdot \mid s_t)) \right) \right] \\
&= \sum_{t=0}^{\infty} \gamma^t \mathbb{E}\left[ R(s_t, a_t) + \alpha \mathcal{H}(\pi(\cdot \mid s_t)) \right] \\
&= Q(s,a) \quad \text{(by soft Q-value definition).}
\end{aligned}
$$

$\square$

### A.2  Convergence of the Optimal Reward

**Corollary A.2.** *For $\rho_\pi = \rho_E$, the optimal reward satisfies $R_Q^*(s,a) = \lambda^{\pi_E} = \lambda^\pi$.*

*Proof.* When $\rho_\pi = \rho_E$, Proposition 4.1 simplifies to

$$R_Q^*(s,a) = \frac{\lambda^{\pi_E} + \lambda^\pi}{2}. \tag{15}$$

Substituting this expression into the loss functions for $\lambda^{\pi_E}$ and $\lambda^\pi$ in Equation (8) yields

$$\min_{\lambda^{\pi_E}} \mathbb{E}_{\rho_E} \left[ \frac{1}{4} (\lambda^\pi - \lambda^{\pi_E})^2 \right], \qquad \min_{\lambda^\pi} \mathbb{E}_{\rho_\pi} \left[ \frac{1}{4} (\lambda^{\pi_E} - \lambda^\pi)^2 \right]. \tag{16}$$

Differentiating each objective with respect to its corresponding target and setting the derivatives to zero gives $\lambda^\pi = \lambda^{\pi_E}$. This result implies that the optimal targets coincide at convergence. Therefore, the optimal reward reduces to

$$R_Q^*(s,a) = \lambda^{\pi_E} = \lambda^\pi. \tag{17}$$

$\square$

## B  Implementation Details

**MuJoCo Suite.** We evaluate RIZE on five Gym (Brockman et al., 2016) MuJoCo (Todorov et al., 2012) locomotion tasks: `HalfCheetah-v2`, `Walker2d-v2`, `Ant-v2`, `Humanoid-v2`, and `Hopper-v2`. Expert trajectories for these tasks are taken from IQ-Learn (Garg et al., 2021) and were generated with Soft Actor–Critic (Haarnoja et al., 2018); each trajectory contains 1,000 state–action transitions. Episode returns are normalized by expert performance with the following expert evaluation returns: `HalfCheetah` (5,100), `Walker2d` (5,200), `Ant` (4,700), `Humanoid` (5,300), and `Hopper` (3,500).

**Adroit.** For `Hammer-v1` from the Adroit suite (Rajeswaran et al., 2018), we use the D4RL dataset (Fu et al., 2021) and filter the top 100 episodes from the original 5,000. The resulting expert subset has an average

return of 16,800, and each episode comprises 200 state–action pairs. We normalize returns in this domain by the average expert return of the selected subset.

**Baselines.** We evaluate five baselines: IQ-Learn (Garg et al., 2021), LSIQ (Al-Hafez et al., 2023), SQIL (Reddy et al., 2020), CSIL (Watson et al., 2023), and Behavior Cloning (BC) (Pomerleau, 1991). For MuJoCo tasks, we use the authors' original code and configurations for IQ-Learn, LSIQ, and CSIL; SQIL is implemented using the LSIQ codebase, and BC follows the CSIL implementation. For `Hammer-v1`, we use the original CSIL code and configs. For IQ-Learn on Hammer, we observe that it does not solve the task even after a small sweep over the entropy coefficient in $\{0.01, 0.03, 0.1\}$ and the loss mode in $\{\texttt{value}, \texttt{v0}\}$; we report $\alpha=0.03$ and `loss=value`. For LSIQ on Hammer, we search over $\alpha \in \{0.01, 0.05, 0.1\}$ and `loss` $\in \{\texttt{value}, \texttt{v0}\}$ and select $\alpha=0.1$, `loss=value` as the better-performing variant. For SQIL, we set $\alpha=0.2$ consistent with our other tasks.

Our architecture integrates components from **Distributional SAC (DSAC)**[1] (Ma et al., 2020) and **IQ-Learn**[2] (Garg et al., 2021), with hyperparameters tuned through search and ablation studies. Key configurations for experiments involving three and ten demonstrations are summarized in Table 1. All implementation details used in our experiments are publicly available at `https://github.com/adibka/RIZE`.

**Distributional SAC Components.** The critic network is implemented as a three-layer multilayer perceptron (MLP) with 256 units per layer, trained using a learning rate of $3 \times 10^{-4}$. The policy network is a four-layer MLP, also with 256 units per layer. To enhance training stability, we employ a target policy—a delayed version of the online policy—and sample next-state actions from this module. For return distribution training $Z_{\phi,\tau}^{\pi}$, we adopt the Implicit Quantile Networks (IQN) (Dabney et al., 2018a) approach by sampling quantile fractions $\tau$ uniformly from $\mathcal{U}(0,1)$. Additionally, dual critic networks with delayed updates are used, which empirically improve training stability. We use the following settings across tasks: replay buffer size $10^6$, batch size 256, 24 quantile levels, and 10,000 pretraining steps. We evaluate every $10^4$ steps, which takes ~3.5 minutes.

**IQ-Learn Adaptations.** Key adaptations from IQ-Learn include adjustments to the regularizer coefficient $c$ and entropy coefficient $\alpha$. Specifically, for the regularizer coefficient $c$, we find that $c = 0.5$ yields robust performance on the `Humanoid` task, while $c = 0.1$ works better for other tasks. For the entropy coefficient $\alpha$, smaller values lead to more stable training. Unlike RL, where exploration is crucial, imitation learning relies less on entropy due to the availability of expert data. Across all tasks, we set initial target reward parameters as $\lambda^{\pi_E} = 10$ and $\lambda^{\pi} = 5$. Furthermore, we observe that lower learning rates for target rewards improve overall learning performance.

Previous implicit reward methods such as IQLearn, ValueDICE, and LSIQ[3] have employed distinct modifications to the loss function. In our setup, two main loss variants are defined:

- *value loss:*

$$\mathcal{L}(\pi, Q) = \mathbb{E}_{\rho_E}[Q(s,a) - \gamma V(s')] - \mathbb{E}_{\rho}[V(s) - \gamma V(s')] - c\,\Gamma(R_Q, \lambda)$$

- *v0 loss:*

$$\mathcal{L}(\pi, Q) = \mathbb{E}_{\rho_E}[Q(s,a) - \gamma V(s')] - (1-\gamma)\mathbb{E}_{p_0}[V(s_0)] - c\,\Gamma(R_Q, \lambda)$$

Here, $\rho$ is a mixture distribution, $p_0$ denotes the initial distribution, $R_Q(s,a)$ is the implicit reward defined as $R_Q(s,a) = Q(s,a) - \gamma V(s')$, the state-value function is given by $V(s') = Q(s',a') - \alpha \log \pi(a'|s')$, and lastly, our convex regularizer is expressed as $\Gamma(R_Q, \lambda) = \mathbb{E}_{\rho_E}[(R_Q - \lambda^{\pi_E})^2] + \mathbb{E}_{\rho_\pi}[(R_Q - \lambda^\pi)^2]$.

The choice between `v0` or `value` loss variants depends on environment complexity: we find that for a complex task like `Humanoid-v2`, the `v0` variant demonstrates greater robustness. And, for `HalfCheetah-v2`, `Walker2d-v2`, `Hopper-v2`, `Ant-v2`, and `Hammer-v1`, the `value` variant performs better.

---

[1] `https://github.com/xtma/dsac`
[2] `https://github.com/Div99/IQ-Learn`
[3] `https://github.com/robfiras/ls-iq/tree/main`

Table 1: Hyperparameters for 3 and 10 Demonstrations (merged where identical)

| Environment | $\alpha$ (3 / 10) | $c$ | lr $\pi$ | lr $\lambda^{\pi_E}$ | lr $\lambda^{\pi}$ (3 / 10) |
|---|---|---|---|---|---|
| Ant-v2 | 0.05 / 0.10 | 0.1 | $5 \times 10^{-5}$ | $1 \times 10^{-4}$ | $1 \times 10^{-5}$ / $1 \times 10^{-4}$ |
| HalfCheetah-v2 | 0.05 / 0.10 | 0.1 | $5 \times 10^{-5}$ | $1 \times 10^{-4}$ | $1 \times 10^{-5}$ / $1 \times 10^{-4}$ |
| Walker2d-v2 | 0.05 / 0.10 | 0.1 | $5 \times 10^{-5}$ | $1 \times 10^{-4}$ | $1 \times 10^{-5}$ / $1 \times 10^{-4}$ |
| Hopper-v2 | 0.20 / 0.20 | 0.1 | $5 \times 10^{-5}$ | $1 \times 10^{-4}$ | $1 \times 10^{-4}$ / $1 \times 10^{-4}$ |
| Humanoid-v2 | 0.05 / 0.10 | 0.5 | $1 \times 10^{-5}$ | $1 \times 10^{-4}$ | $5 \times 10^{-5}$ / $1 \times 10^{-5}$ |
| AdroitHandHammer-v1 | 0.30 / 0.30 | 0.1 | $3 \times 10^{-5}$ | $1 \times 10^{-4}$ | $5 \times 10^{-5}$ / $5 \times 10^{-5}$ |

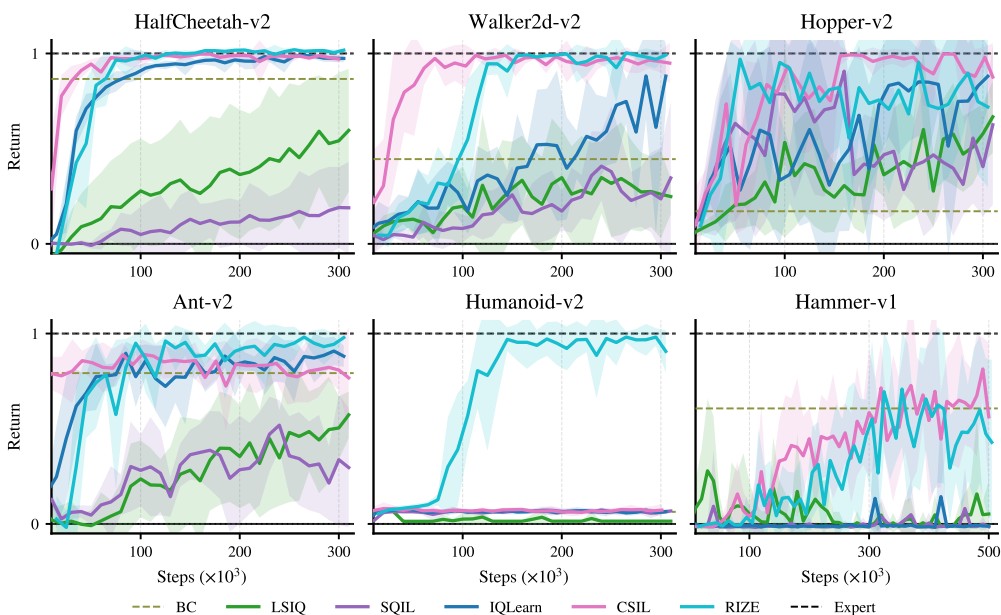

Figure 6: Learning curves on MuJoCo and Adroit tasks with 3 expert demonstrations. Lines show the mean normalized return across five seeds; shaded regions denote 95% confidence intervals.

## C   Additional Experiments

This section augments the main experimental results with additional experiments, including targeted ablations (critic architecture, regularization design, and loss choice) and a hyperparameter sensitivity analysis.

### C.1   Main Results (Extended)

Figure 6 complements the main-text results by presenting learning curves for the three-demonstration regime. Learning curves mirror the 10-demo setting (cf. Figure 3): RIZE and CSIL performs better than the baselines with more stable learning across tasks; CSIL shows strong early progress due to behavior-cloning initialization; IQ-Learn is competitive yet below RIZE; and LSIQ/SQIL trail behind. Task-wise, `Humanoid-v2` remains solved only by RIZE, while on `Hammer-v1` RIZE and CSIL lead. As expected with fewer demonstrations, normalized returns are lower and variance is higher, but the relative ordering of methods and the sample-efficiency patterns are consistent with the RLiable aggregates reported in the main body (Figure 1).

### C.2   Recovered Reward (Extended)

We complement the main-text analysis with additional plots in the 3–demonstration regime and cross-method comparisons. First, Figure 7 mirrors the main-body analysis for the 10–demo setting: it overlays recovered

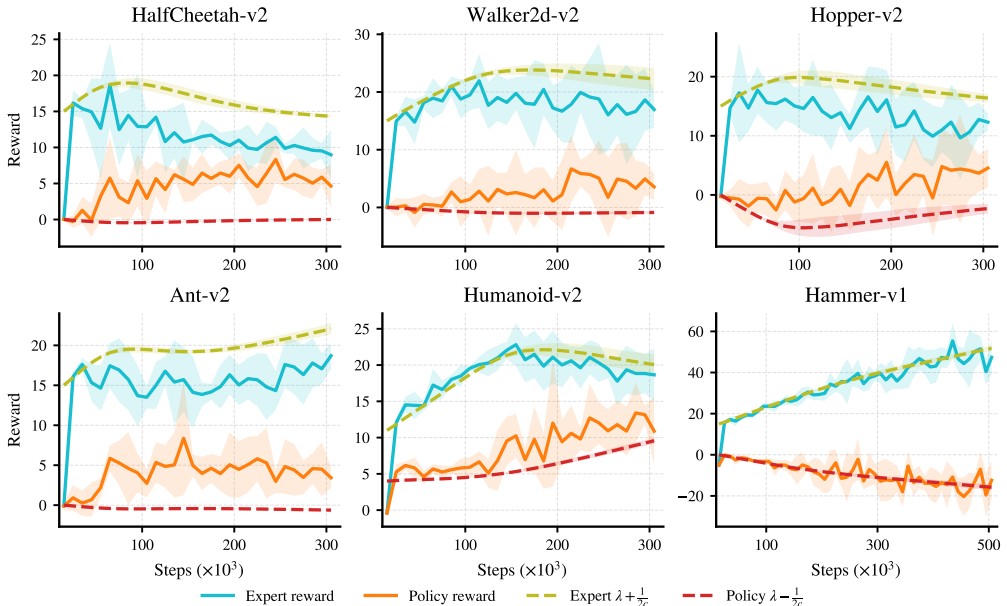

Figure 7: Implicit reward curves for expert and policy samples on MuJoCo and Adroit tasks with 3 expert demonstrations. Each subplot reports the mean across five seeds, with shaded regions showing the 95% confidence interval. Theoretical upper and lower bounds derived in this work are overlaid as separate curves in each subplot.

reward trajectories (expert and policy) with the theoretical bounds implied by our regularizer $\Gamma(R_Q, \lambda)$ and adaptive targets (Corollary 4.2). As in the main text, the curves remain within the predicted band, and the band's position/width adapts by task, reflecting how the adaptive targets track and limit expert/policy rewards in practice.

Across all three figures—Figures 9, 8, and 10a—RIZE and LSIQ keep implicit rewards bounded, whereas SQIL and IQ-Learn exhibit drifting or unbounded rewards on the challenging `Humanoid-v2` and `Hammer-v1` tasks. In LSIQ, boundedness is likely due to its target clipping; notably, expert-sample rewards concentrate near zero. By contrast, SQIL fails to control reward scales on nearly all tasks, and IQ-Learn is particularly unstable on `Humanoid-v2` and `Hammer-v1`. These observations support the view that adaptive, task-sensitive regularization is effective in maintaining calibrated implicit rewards.

### C.3 Critic Value-estimation

We examine how critic estimates evolve in RIZE versus baselines. Unlike methods that train point-estimate $Q$-networks, RIZE employs an IQN critic that models the full return distribution $Z(s, a)$ and optimizes losses using its expectation $\mathbb{E}[Z(s, a)]$ (Dabney et al., 2018a). Because all policies are updated by maximizing state–action values, unbounded estimates can destabilize training and undermine robustness.

Figures 10b, 11, and 12 show that RIZE and LSIQ maintain bounded values on the challenging `Humanoid-v2` and `Hammer-v1` tasks, whereas SQIL and IQ-Learn exhibit large, drifting estimates. For LSIQ, boundedness primarily arises from target clipping to $[-200, 200]$, which also explains why expert-sample values cluster near $+200$. Importantly, bounded critics alone do not guarantee expert-level control within our training budgets (3e5 steps for MuJoCo; 5e5 for Adroit): LSIQ remains below RIZE in Figures 3 and 6, indicating that value stability achieved by adaptive reward regularization can translate into better performance than rigid critic clipping with fixed reward targets.

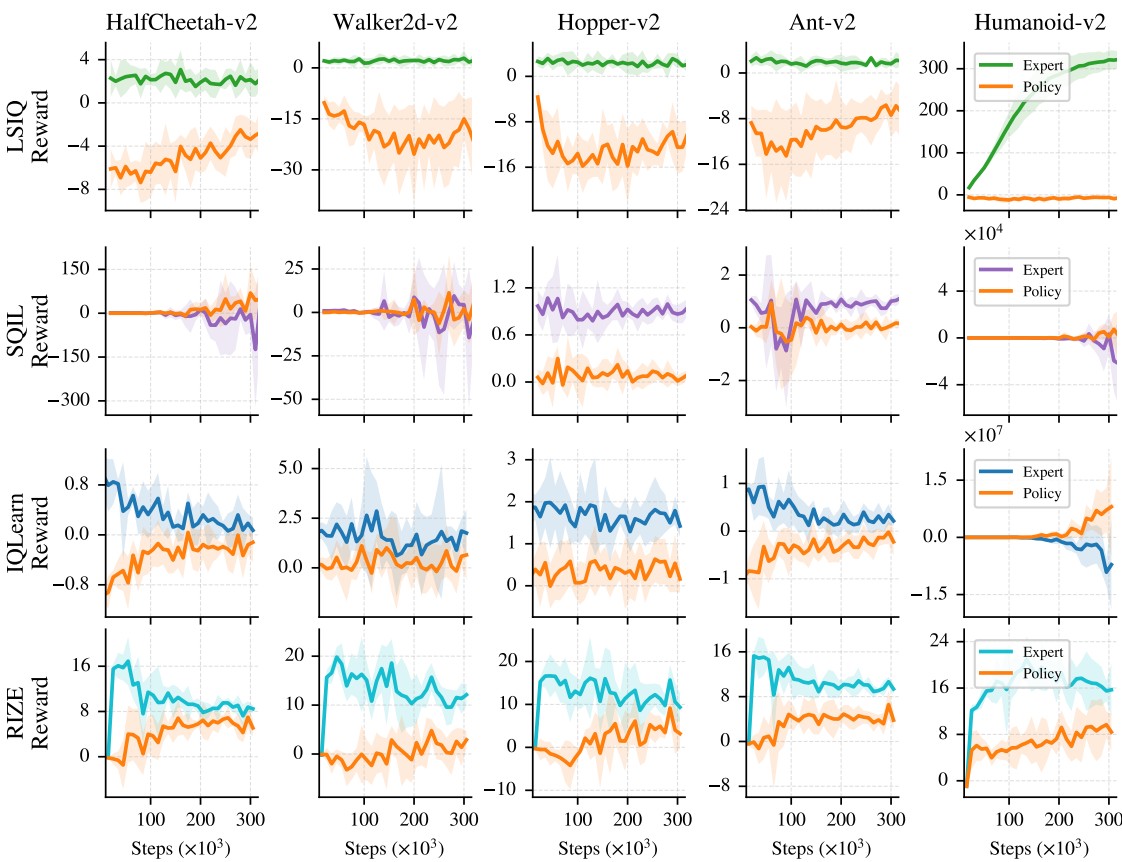

Figure 8: Implicit reward curves on MuJoCo tasks comparing RIZE with SQIL, IQ-Learn, and LSIQ using 10 expert demonstrations. Lines show the mean across five seeds; shaded regions denote 95% confidence intervals.

### C.4 Ablation on Critic Architecture (Extended)

Figure 13 complements the main-body ablation by examining *recovered rewards with theoretical bounds* when RIZE uses a point-estimate $Q(s, a)$ critic in place of the IQN return-distribution critic $Z(s, a)$ (cf. Figure 5 for returns, and Figure 7 for bounded rewards under IQN). With the $Q$ critic, implicit rewards are less well constrained in `Humanoid-v2` and `Hammer-v1` environments: we observe more excursions beyond the predicted band and larger oscillations, in contrast to the IQN setting where rewards remain tightly within the adaptive interval. This comparison suggests that both components contribute to reward stability: the squared-TD regularizer with adaptive targets provides principled bounds, and modeling the return distribution with IQN supplies steadier targets/updates that help keep rewards within those bounds. Together, these effects yield more reliable policy learning and align with the performance gap observed in the return curves.

### C.5 Ablation on Loss Choice (HL-Gauss in LSIQ)

Recent work argues that some of the gains attributed to distributional RL may stem from the *loss* rather than from modeling the return distribution itself (Farebrother et al., 2024; Ayoub et al., 2024). Following this "loss swapping" view, we replace the mean-squared error (MSE) used in LSIQ's critic with the HL-Gauss classification loss (Farebrother et al., 2024), which discretizes the value range into bins and trains with a Gaussian-smoothed target over bins (turning value regression into calibrated classification).

**Setup.** Among our baselines, LSIQ (and SQIL) are natural candidates because their critics use MSE-like objectives, whereas IQ-Learn and RIZE optimize MaxEnt-style IRL objectives. We therefore apply the swap

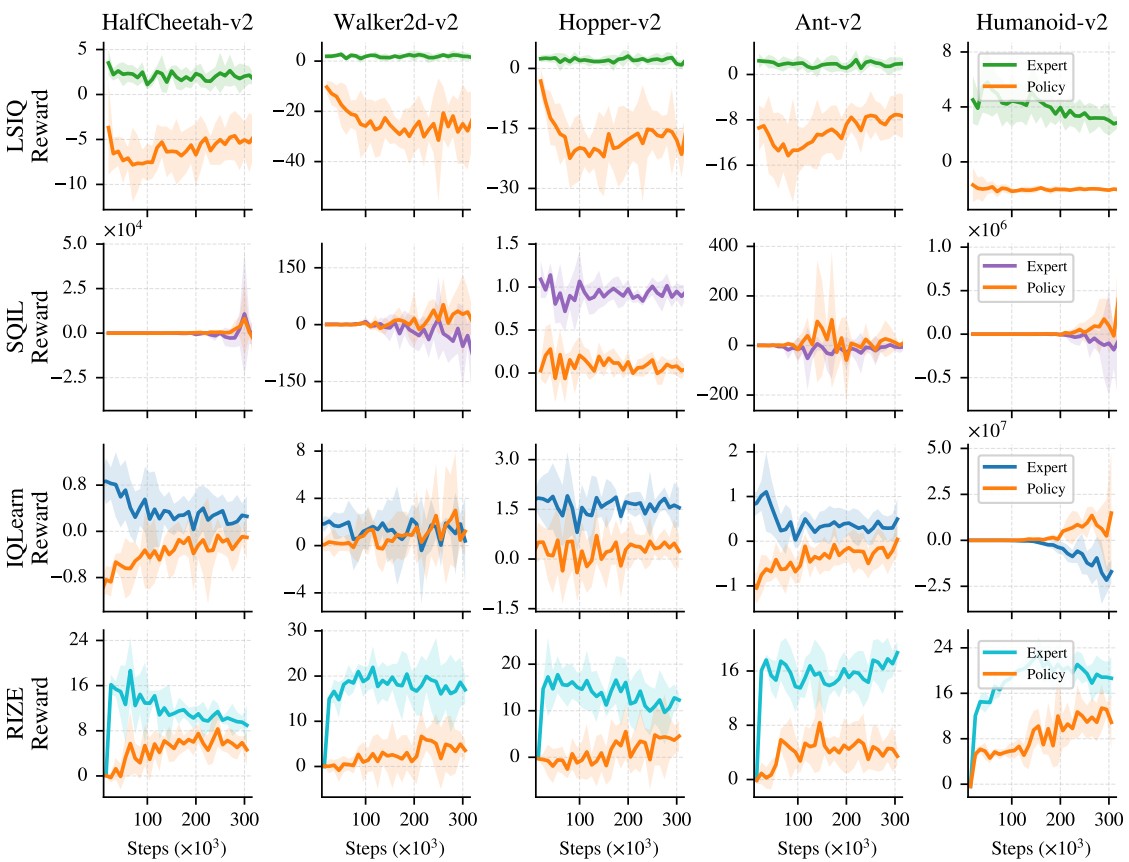

Figure 9: Implicit reward curves on MuJoCo tasks comparing RIZE with SQIL, IQ-Learn, and LSIQ using three expert demonstrations. Lines show the mean across five seeds; shaded regions denote 95% confidence intervals.

to LSIQ and keep all other components unchanged. Following LSIQ's critic clipping in $[-200, 200]$, we set $v_{\min} = -200$, $v_{\max} = 200$, use `num_bins`=101, and Gaussian width $\sigma = 8$. We evaluate on `Walker2d-v2`, `Ant-v2`, and `Hammer-v1` with three demonstrations.

**Results.** As shown in Figure 14, HL-Gauss underperforms the original MSE-based LSIQ on all three tasks: it yields no improvement on `Walker2d-v2` or `Ant-v2`, and remains near zero on `Hammer-v1`. A likely cause is structural: LSIQ's critic objective is the sum of two squared-error terms (expert and policy), which is integral to its least-squares/mixture formulation; swapping those squared losses for a categorical (multi-bin) classification surrogate alters the optimization geometry and weakens the intended expert–policy mixture shaping. In contrast, standard Q-learning—where loss swaps have shown benefits—regresses to a single TD target, making the classification surrogate a closer drop-in for MSE.

### C.6 Ablations on Regularization Strategies

We study how the regularizer design affects performance on `Walker2d-v2`, `Ant-v2`, and `Hammer-v1` with three demonstrations. Our baseline uses a squared TD-error regularizer $\Gamma$ in Equation (7) with separately optimized adaptive targets for expert and policy samples. We compare this to two alternatives: (i) a *coupled* target, where we replace the separate targets with a single shared $\lambda$ (initialized to 0 or 10), and (ii) a plain L2-regularizer on rewards (no targets), akin to IQ-Learn's constraint.

Figure 15 shows that squared TD with *separate, adaptive* targets yields the most robust and highest returns across tasks. Using a coupled target is brittle: $\lambda$=10 attains expert-like performance on `Ant-v2`, remains

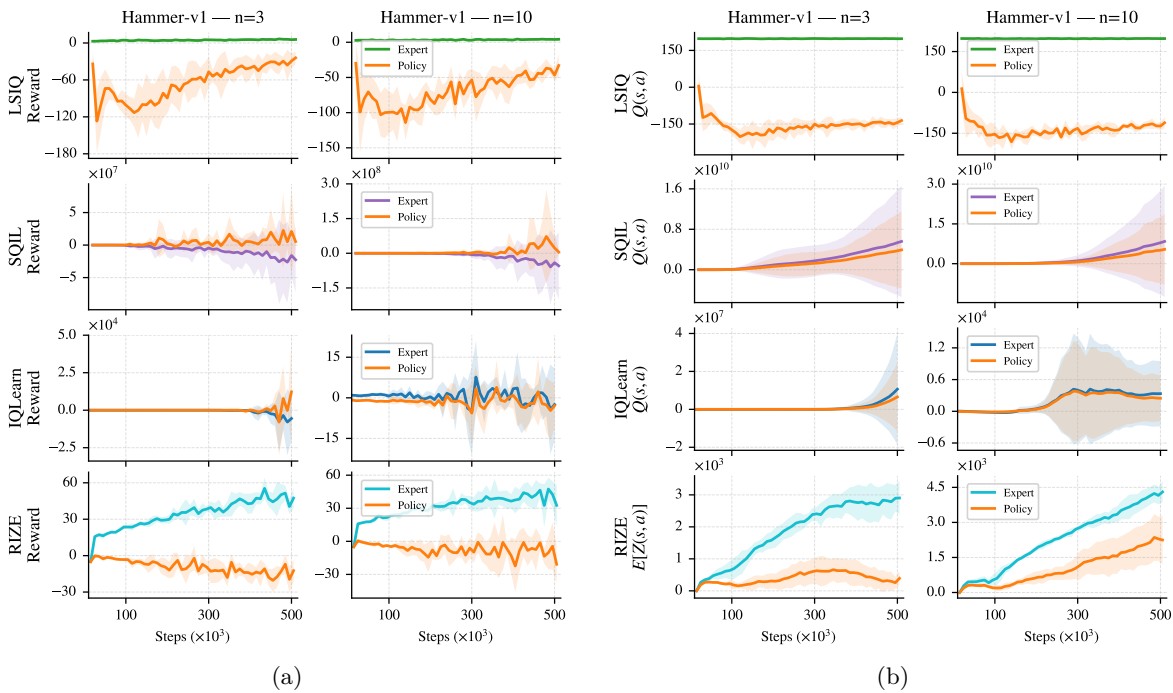

Figure 10: Adroit Hammer-v1 results: **(a)** implicit reward curves and **(b)** value–estimation curves, each reported for 3 and 10 expert demonstrations (n denotes number of trajectories). Lines show the mean across five seeds; shaded regions denote 95% confidence intervals.

below the original result on `Walker2d-v2`, and fails on `Hammer-v1`, while $\lambda=0$ collapses across all tasks. Replacing squared TD with a plain L2 penalty also fails on all three tasks in our setup. We attribute this difference from IQ-Learn's reports to implementation and hyperparameter choices tailored to our setting: RIZE employs an IQN critic, target policy networks, different learning rates (notably for the policy), a higher entropy coefficient (0.05 vs. 0.01), and a smaller regularizer coefficient (0.1 vs. 0.5). These choices were tuned for squared TD with adaptive targets; swapping to an L2 penalty disrupts learning. Overall, the results indicate that *adaptive, decoupled targets* are crucial for stabilizing reward across tasks.

## C.7   Hyperparameter Tuning

We present our analysis and comparison of important hyperparameters utilized in our algorithm. Plots depict mean over five seeds with 95% confidence intervals.

**Adaptive Targets.**   Selecting appropriate initial values and learning rates for the automatic fine-tuning of $\lambda^{\pi_E}$ and $\lambda^{\pi}$ is critical in our approach. First, we observe that a suitable learning rate is essential for the stable training of our imitation learning agent, as illustrated in Figure 16a. Our findings indicate that $\lambda^{\pi}$ must be optimized very slowly; using larger learning rates can destabilize training and hinder progress. In contrast, $\lambda^{\pi_E}$ demonstrates greater resilience when optimized with higher learning rates. Additionally, $\lambda^{\pi_E}$ remains robust even with varying initial values. However, as shown in Figure 16b, failing to select an appropriate initial value for $\lambda^{\pi}$ can negatively impact learning. Overall, Figures 16a and 16b highlight the need for careful selection of both the learning rate and initial value when optimizing $\lambda^{\pi}$, while $\lambda^{\pi_E}$ exhibits considerable robustness in this regard.

**Regularization Coefficient.**   Our experiments on the regularizer coefficient $c$ reveal that smaller values of $c$ encourage expert-like performance, while larger values overly constrain rewards and targets, limiting learning. This finding highlights the critical role of selecting an appropriate $c$, as it directly impacts the

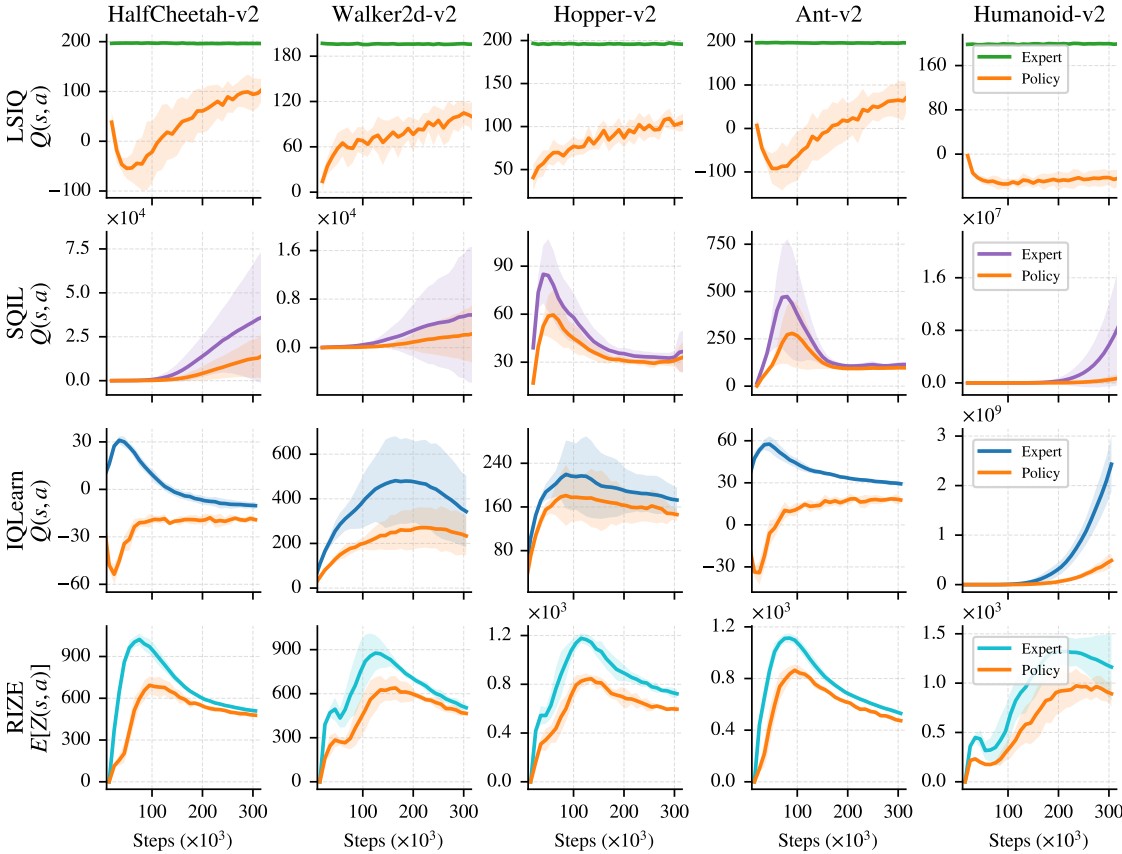

Figure 11: Value-estimation curves on MuJoCo tasks comparing RIZE (IQN critic (Dabney et al., 2018a)) with SQIL, IQ-Learn, and LSIQ (point-estimate $Q$) using 10 expert demonstrations. Lines are means over five seeds; shaded regions denote 95% confidence intervals.

balance between learning from expert data and regularization: higher values prioritize regularization at the cost of learning, whereas smaller values favor learning but reduce regularization (see Figure 17a).

**Entropy Coefficient.** We observe that the entropy coefficient is a crucial hyperparameter in inverse reinforcement learning (IRL) problems. As shown in Figure 17b, IRL methods typically require small values for $\alpha$, a point previously noted (Garg et al., 2021). With expert demonstrations available, an imitation learning (IL) policy does not need to explore for optimal actions, as these are provided by the demonstrations. Consequently, higher values of $\alpha$ can lead to training instability, ultimately resulting in policy collapse.

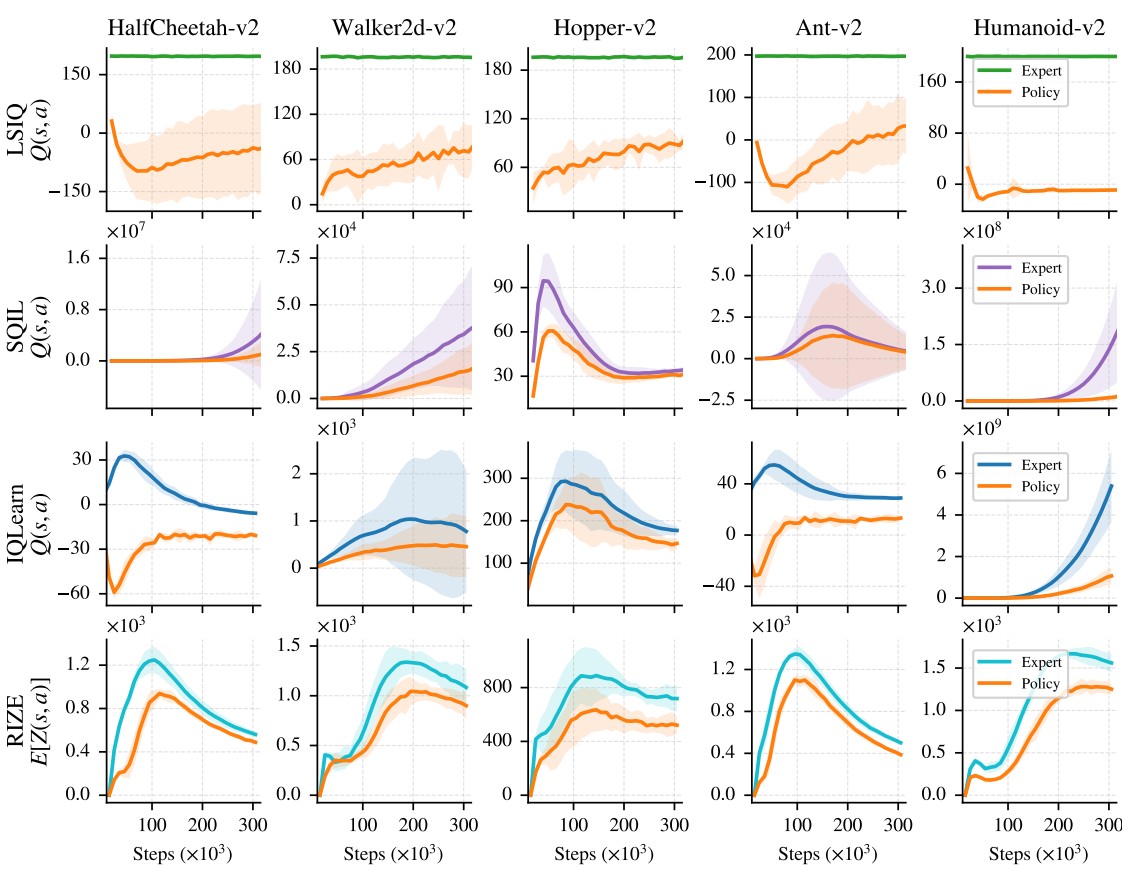

Figure 12: Value-estimation curves on MuJoCo tasks comparing RIZE (IQN critic (Dabney et al., 2018a)) with SQIL, IQ-Learn, and LSIQ (point-estimate $Q$) using three expert demonstrations. Lines are means over five seeds; shaded regions denote 95% confidence intervals.

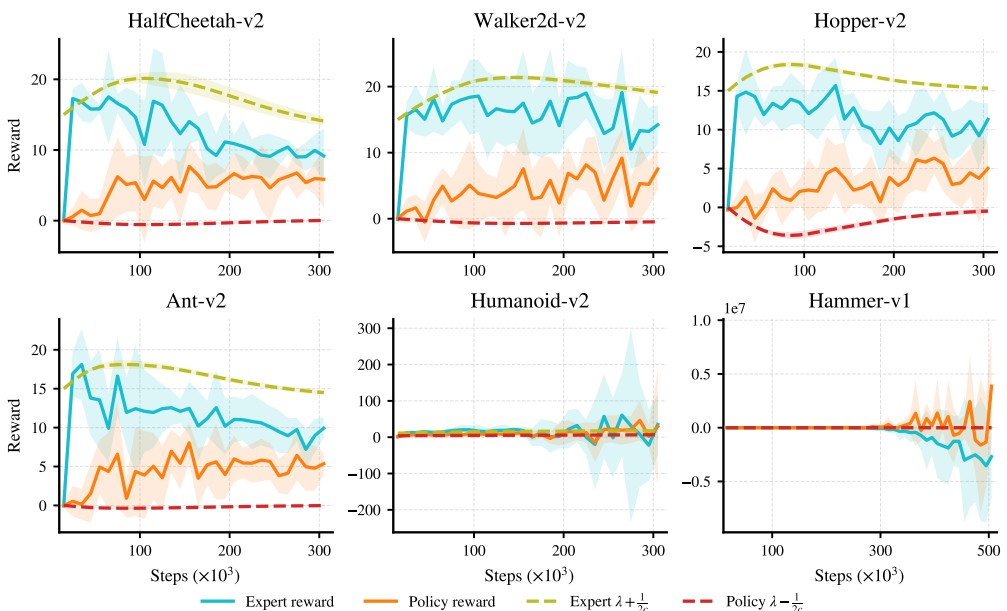

Figure 13: Implicit reward curves for expert and policy samples on MuJoCo and Adroit tasks using RIZE with a classic $Q(s, a)$ critic. Each subplot shows the mean over five seeds with shaded 95% confidence intervals, using three expert demonstrations. Theoretical upper and lower bounds derived in this work are overlaid as separate curves in each subplot.

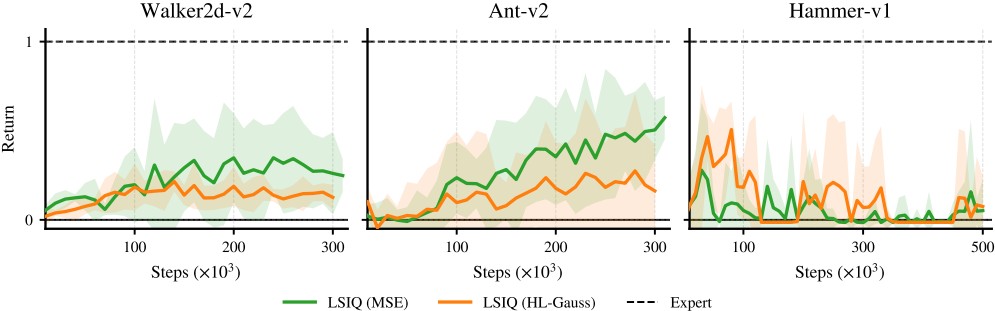

Figure 14: Loss swap in LSIQ: MSE vs. HL-Gauss on `Walker2d-v2`, `Ant-v2`, and `Hammer-v1` with three demonstrations. We keep $v_{\min} = -200$, $v_{\max} = 200$, `num_bins`=101, and $\sigma = 8$. HL-Gauss does not improve over MSE and remains near zero on `Hammer-v1`.

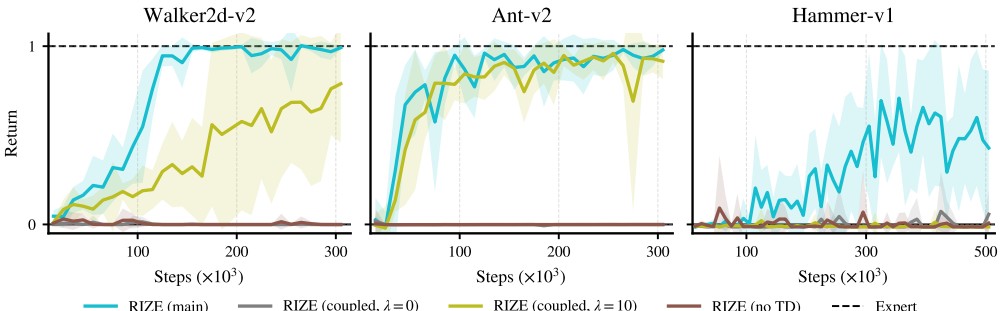

Figure 15: Ablation of regularization strategies on selected tasks (Walker2d, Ant, Hammer) using three expert demonstrations. Results are expert-normalized and reported as the mean over five seeds with 95% confidence intervals.

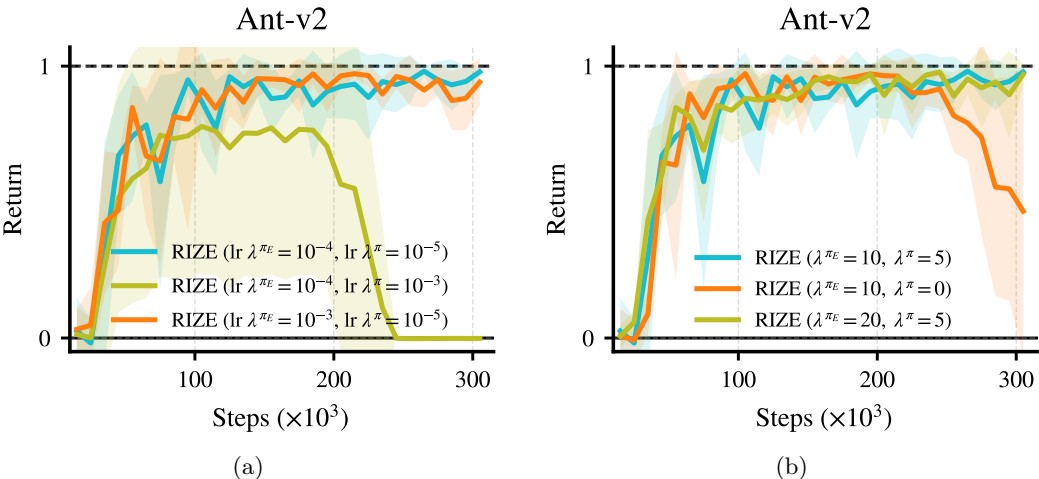

Figure 16: Fine-tuning analysis of adaptive targets. **(a)** Learning rates: Turquoise represents our method's primary result with learning rates of 1e−4 for $\lambda^{\pi_E}$ and 1e−5 for $\lambda^{\pi}$. Orange and blue lines indicate higher learning rates (e.g., 1e−3) for $\lambda^{\pi_E}$ and $\lambda^{\pi}$, respectively. **(b)** Starting values: Turquoise shows the main result with initial values of 10 for $\lambda^{\pi_E}$ and 5 for $\lambda^{\pi}$, while other lines explore different starting values. Three trajectories are used throughout the analysis.

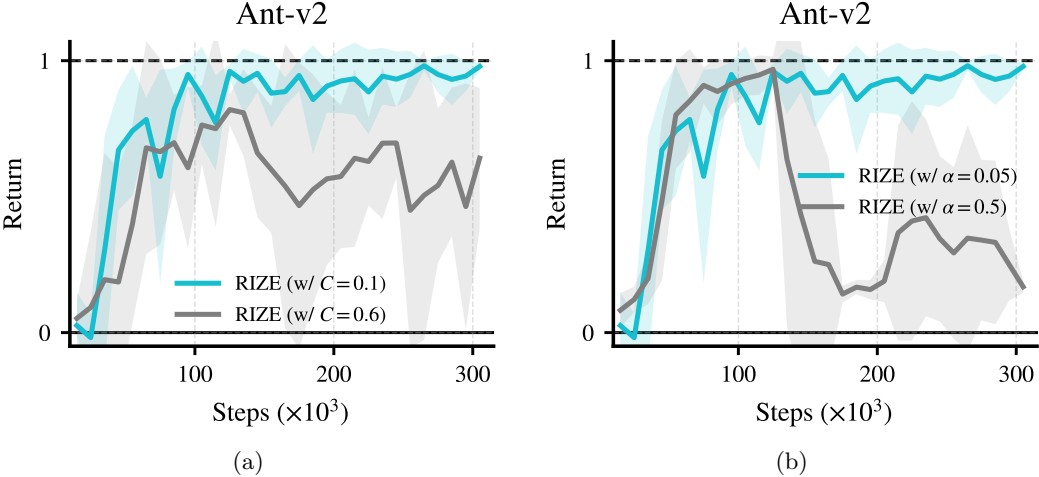

Figure 17: **(a)** Effect of the regularizer coefficient $c$. Turquoise shows the primary result of our method with $c = 0.1$, while gray represents a larger value ($c = 0.6$). **(b)** Effect of the temperature parameter $\alpha$. Turquoise shows the result with $\alpha = 0.05$, and gray corresponds to a larger value ($\alpha = 0.5$). Three trajectories are used throughout the analysis.

