# OpenReview forum: "RIZE: Adaptive Regularization for Imitation Learning"
_TMLR — Accepted by TMLR_

### Review · Reviewer_q66R · 2025-08-29

**Summary Of Contributions:**

This paper tackles the problem of inverse reinforcement learning (RL), i.e. learning a policy given only trajectories and their returns.
The authors argue that previous methods in this area, which only considered point-estimate Q-value functions, do not capture the full information and are thus slow to converge.

In this work, the authors leverage the distributional Q-value functions, which capture the inherent aleatoric uncertainty in returns under stochastic policies. They theoretically showed that the reward functions learned by such functions are bounded and thus promote stable optimization. The resulting practical algorithm, RIZE, is shown to be able to learn well-performing policies (in terms of returns) with only a few trajectories to learn from.

Overall, the manuscript is well-written. The presented results are compelling, albeit hard to judge their significance.

**Audience:**

Yes

**Audience Explanation:**

The general insight of incorporating various sources (in this case, aleatoric) uncertainty for improving decision-making and optimization is useful.

**Broader Impact Concerns:**

Doesn't seem to be applicable.

**Claims And Evidence:**

Yes

**Claims Explanation:**

The main claim of this paper is that using value functions that take (aleatoric) uncertainty into account is beneficial for inverse RL. The authors showed compelling evidence:

1. In Prop. 4.1, the authors analyze the impact of such a decision on the (implicit) reward function learned from expert demonstrations/trajectories. They showed that this reward function is bounded, and thus makes optimization stable
2. In the empirical analysis (Sec. 5), the authors provided empirical validation for the previous point, along with an ablation study.

**Requested Changes:**

Some comments to strengthen the submission:

1. I believe it would be very useful if the authors discuss at a high level the intuition on why considering aleatoric uncertainty in the Q-value function leads to a bounded reward function. While I could see that considering uncertainty, i.e., having more information, is generally useful for various tasks like optimization and learning, the connection between uncertainty and boundedness is unintuitive to me.
2. I would suggest the authors provide a significance test on top of the empirical plots. Eyeballing plots is unreliable for judging significance.
3. I suggest the authors open-source their implementation.

---

### Review · Reviewer_h3JC · 2025-09-02

**Summary Of Contributions:**

## Summary

This paper introduces an online imitation learning method, called RIZE,
which extends the IQLearn method by introducing the following
improvements:

1.  Substituting the Q-function parameterization with a distributional
    variant, and using the induced mean in place of the Q-function; and
2.  An adaptive reward regularization scheme, which regularizes rewards
    toward learned "centers" over the course of training.

Through experiments in Mujoco tasks, RIZE considerably outperforms the
tested baselines (including IQLearn). Notably, in the Humanoid task,
RIZE approximately recovers expert performance, while the baseline
methods fail entirely.

## Strengths

The proposed method is simple and easy to implement, while achieving
strong performance. The adaptive regularization scheme is novel and
interesting. The authors carry out a few ablations to justify the
necessity of the novel pieces of RIZE for achieving strong performance.

## Weaknesses

The modifications made to IQLearn are not very well motivated.
Particularly:

- **Distributional RL**. The only motivation really appears to be that
  modeling return distributions is successful in RL, so it should be
  successful in imitation learning. But in principle, there is no clear
  reason for this to be the case, and as I discuss below, there are
  confounding factors that might explain the performance achieved by
  incoroporating distributional RL in this framework.
- **Adaptive reward regularization**. This is a seemingly novel
  technique, but again it is not clear why it should be expected to
  work. Any theoretical results about this scheme would hold for the
  case where the regularization parameters are zero (which is the case
  of IQLearn)—thus, these results do not actually suggest that learning
  the regularization is beneficial in principle. In fact, it is not
  clear to me why this doesn't lead to divergence.

Moreover, as I explain below, I have some concerns with the experimental
procedure, as well as the breadth of the experiments, baselines, and
ablations. Consequently, it is difficult for me to fully appreciate both
the performance of the proposed method relative to the current state of
the field and the effects of the individual components that were
introduced.

**Additional Comments:**

1.  The motivation for the adaptive reward targets is not at all clear
    to me. Particularly, under this parameterization, is it not
    possible (perhaps even likely) for the $\lambda^{\pi_E}$ and
    $\lambda^\pi$ terms to diverge? It's especially odd to me that
    $\lambda^\pi, \lambda^{\pi_E}$ are "decoupled", did you try
    enforcing $\lambda^\pi \equiv \lambda^{\pi_E}$?
2.  Can you explain why the IQLearn implicit rewards diverge in
    Humanoid? From what I can tell, the objective of IQLearn is
    equivalent to that of RIZE, but with $\lambda^\pi = \lambda^{\pi_E}
       = 0$ (equivalently, $\lambda_{\min} = \lambda_{\max} = 0$).
    So by Corollary A.2, the implicit rewards of IQLearn should also be
    bounded (and even to a smaller range than those of RIZE). So by
    your own argument for RIZE, shouldn't IQLearn at least produce
    conservatively bounded implicit rewards?
3.  In Figure 6, I do not see the curve for RIZE with baseline $L_2$
    regularization. Is it lying on the $x$-axis? This would surprise
    me, given that this formulation appears almost equivalent to
    IQLearn, which performed well in Ant.
4.  I believe the paper places too much attribution to distributional
    RL. The RIZE method effectively only uses distributional RL as an
    empirical trick; the return distributions are used *only* to extract
    expected values, so the distributions themselves are not
    contributing anything explicitly. Thus, it feels like a stretch to
    title the paper "Regularized Imitation Learning via Distributional
    RL". It also removes emphasis from the adaptive regularization
    scheme, which seemingly also plays a large role, and which
    actually changes the learning objective.


## References

1.  "Of Moments and Matching". Swamy et al., 2021. *International
    Conference on Machine Learning.*
2.  "Adversarial Imitation Learning via Boosting". Chang et
    al., 2024. *International Conference on Learning Representations*.
3.  "Non-Adversarial Inverse Reinforcement Learning via Successor
    Feature Matching". Jain et al., 2025. *International Conference
    on Learning Representations*.
4.  "Diffusing States and Matching Scores: A New Framework for
    Imitation Learning". Wu et al., 2025. *International Conference
    on Learning Representations*.
5.  "Switching the Loss Reduces the Cost in Batch Reinforcement
    Learning". Ayoub et al., 2024. *International Conference on Machine
    Learning*.
6.  "Stop Regressing: Training Value Functions via Classification for
    Scalable Deep RL". Farebrother et al., 2024. *International
    Conference on Machine Learning*.

**Audience:**

Yes

**Audience Explanation:**

Online imitation learning is a very active area of research in machine
learning. Despite my concerns about the empirical results, the proposed
method does achieve strong performance in the difficult Humanoid task,
and especially due to its simplicity, I believe the imitation learning
community will find this useful.

**Broader Impact Concerns:**

No broader impact concerns.

**Claims And Evidence:**

No

**Claims Explanation:**

The theoretical results appear to be correct. However, the claims about
the empirical performance are too strong from my perspective. This is
due to the relatively small collection of environments tested relative
to competing works, the lack of recent strong baselines, and the lack of
important ablations (each discussed below).

**Requested Changes:**

## Critical changes

1.  The literature review omits several works on non-adversarial
    approaches to IRL / imitation learning. Off the top of my head,
    [1] and [3-4] from the references I listed in **Additional Comments**
    appear quite relevant.
2.  Regarding the performance of RIZE, while it appears quite good, it
    is tested on relatively few environments/tasks. Additionally, it is
    not compared to some recent baselines which I suspect would perform
    very competitively (see e.g. [2-4] in the references I listed under
    **Additional Comments**).
3.  While I appreciate the ablation in Section 5.3, some questions
    still remain. For instance, [5-6] in the references I listed in
    **Additional Comments** argue that the performance improvements brought
    along by distributional RL should be attributed to the loss
    function, and not the fact that they're modeling the return
    distribution. Thus, perhaps by substituting the Q function MSE in
    any of the tested methods with something like HL-Gauss [6], we
    could see performance enhancements rivaling RIZE.
4.  Figure 1 is difficult to interpret. It says it's plotting the
    return of the "top 25% episodic returns across 5 seeds". How
    exactly is this computed? How many rollouts do you estimate for
    each seed? Is the top 25% taken seed-wise or over all returns
    generated by a given method? Based on the information given, it
    sounds like you're running one rollout per seed and plotting the
    top 25%, which is (I'm guessing?) simply the best return. Also,
    there seems to be a discrepancy between Figure 1 and 2; somehow the
    baselines look much weaker in Figure 1. The results should be
    explained more precisely.
5.  In section 5.4, the ablation should be conducted on the full suite
    of tasks&#x2014;it is not clear that any conclusions drawn from Ant
    generalize.
6.  In section 5.4, I believe there is a crucial ablation missing:
    namely, the setting where $\lambda^\pi = \lambda^{\pi_E}$. The
    "expert-focused TD" and "policy-focused-TD" configurations that you
    test are quite unnatural to me: I don't know why there would be
    such an asymmetry to the regularization method for the reward
    function under the policy's vs the expert's state visitation
    distributions. However, intuitively I would have thought the method
    would make the most sense with $\lambda^\pi = \lambda^{\pi_E}$,
    which still employs a degree of adaptive regularization that you argue for.


## Strengthening changes

1.  The term "value distribution" isn't entirely accurate, "return
    distribution" is preferred. This is because "value" refers to the
    value function which measures expected returns by definition,
    whereas the "return" is random (and its distribution is what is
    modeled by distributional RL).
2.  Towards the end of section 1, it says "complex tasks like *Humanoid*,
    where both adaptive rewards and distributional learning are
    essential". Can you provide a citation for this? Surely
    distributional RL is not essential for humanoid, since many methods
    can solve this task without distributional RL. Also, it is not
    clear yet what you mean by "adaptive rewards".
3.  There are inaccuracies in the discussion of the distributional soft
    Bellman operator. Particularly, under equation (1), you claim that
    the operator exhibits contractions properties and that this yields
    convergence to the optimal distributional value function. The
    discrepancy is the following:
    1.  The operator $\mathcal{B}^\pi_D$ that you define is a policy
        evaluation operator, its iterates do not yield an optimal
        distributional value function.
    2.  The closely-related distributional soft optimality operator is
        not a contraction (so, contraction properties are not
        responsible for convergence to an optimal distributional value function).
    3.  Despite the fact that the distributional soft optimality
        operator has convergent iterates, its iterates do not converge
        to a classically optimal distributional value function; it is
        only optimal for the entropy-regularized RL objective (but this
        is more of a nit).
4.  Just above section 3.3, you refer to "the value distribution
    $Z_\tau(s, a)$", but this is not a distribution&#x2014;for any
    $\tau$, $Z_\tau(s, a)$ is a scalar.
5.  Since Corollary A.2 is framed (at least from my perspective) as a
    key result, it should be in the main text.

---

### Review · Reviewer_uutq · 2025-09-17

**Summary Of Contributions:**

This paper introduces RIZE, a novel imitation learning method that extends the Maximum Entropy IRL framework with two key innovations:
- Adaptive reward targets for squared TD regularization: Unlike prior implicit reward methods (e.g., SQIL, LSIQ) that fix reward targets to $\pm 1$, RIZE learns adaptive targets $\lambda^{\pi_E}, \lambda^{\pi}$ that evolve during training. This provides flexible, context-sensitive reward alignment and provably bounds the implicit reward within a well-defined interval.
- Integration of distributional reinforcement learning: The authors incorporate distributional critics (via quantile-based value networks) into IRL. They optimize policies with respect to the expectation of the learned return distribution, capturing richer uncertainty while stabilizing learning.

The authors present a theoretical analysis of reward boundedness (Proposition 4.1, Corollary A.2--A.3), a practical algorithm built on distributional SAC, and extensive experiments on four MuJoCo benchmarks. RIZE consistently outperforms SQIL, LSIQ, and IQ-Learn, and uniquely achieves expert-level performance on the challenging Humanoid-v2 task with only three expert demonstrations. Ablations examine the role of adaptive regularization, distributional critics, and hyperparameter sensitivity.

**Additional Comments:**

- The writing is clear, and the introduction and related work position the contribution well.
- The theoretical analysis is concise and correctly caveated.
- Figures are generally informative, though including raw curves and confidence intervals would improve interpretability.
- Table 1 is helpful, but reproducibility would benefit greatly from code release.

**Audience:**

Yes

**Audience Explanation:**

Yes. The work addresses fundamental challenges in imitation learning — robust reward recovery and stability under limited expert data. Both adaptive regularization and distributional critics are conceptually novel in the context of non-adversarial IRL, and the empirical demonstration on Humanoid with only three demonstrations is particularly appealing. TMLR’s audience, which includes researchers in reinforcement learning, imitation learning, and robotics, would be interested in these findings.

**Broader Impact Concerns:**

The broader impact is generally positive. RIZE aims to improve robustness and efficiency of imitation learning, which is directly relevant to robotics and autonomous systems. Potential concerns include:
- Reward mis-specification risks: While adaptive bounds improve stability, learned rewards could still encode biases in expert demonstrations.
- Safety-critical domains: Using adaptive implicit rewards without strong convergence guarantees could pose risks in safety-sensitive applications (e.g., healthcare, autonomous driving).

These concerns should be acknowledged, but none appear to be critical blockers.

**Claims And Evidence:**

Yes

**Claims Explanation:**

- Theoretical claims: The paper provides clear derivations showing that the adaptive TD regularizer constrains rewards within bounded intervals. The proofs in Appendix A are correct and easy to follow. The authors are careful to state that convergence guarantees for alternating updates remain open, which appropriately limits their theoretical claims.

- Empirical claims: The main claim — that RIZE improves robustness and achieves expert-level Humanoid performance with few demonstrations — is supported by strong results. Ablations show that both adaptive regularization and distributional critics matter. However:
    - The evaluation metric “top 25% returns” is non-standard. While it highlights convergence, conventional practice (mean or median returns with confidence intervals, AUC, final performance) would be a stronger basis for claims of efficiency and stability.
    - Hyperparameter tuning fairness across baselines, particularly in the 3-demo setting, is not fully transparent. RIZE appears to receive environment-specific tuning and alternative losses (value vs.\ v0), whereas it is unclear if baselines were given comparable treatment.

Overall, the evidence is convincing for RIZE’s effectiveness, but additional results and clarifications would further strengthen the claims.

**Requested Changes:**

- Provide conventional learning curves (mean $\pm$ CI over seeds/episodes), final-100-episode means, and AUC, alongside the “top-25%” metric currently emphasized.
- Clarify how tuning budgets and search spaces were matched across RIZE and baselines, especially in the 3-demo regime.
- Isolate the contributions of (a) quantile-based critics vs.\ point-estimate critics, (b) number of quantiles, and (c) stabilizers like target-policy and double critic.
- Provide time-series plots of $\lambda^{\pi}, \lambda^{\pi_E}$ and histograms of recovered rewards to confirm they respect the theoretical bounds.
- Add Behavioral Cloning (with and without fine-tuning) as a low-data reference, and optionally include an adversarial baseline (e.g., GAIL or DAC) on at least one environment.
- Expand the appendix with implementation details such as replay-buffer size, batch size, target-network update rate $\tau$, quantile samples per update, warm-up steps, evaluation frequency, and wall-clock compute.

---

### Author Response · Authors · 2025-10-04

@~Aleksandra_Faust1, @uutq, @h3JC, @q66R:

We have uploaded a revised PDF and updated code (v2 in supplementary materials) addressing the points in our prior rebuttal (posted 29 Sept 2025). Key changes include expanded experiments (new tasks/baselines), ablations (e.g., quantile critics, coupled λ), clarified metrics/visuals, and theoretical fixes.

We appreciate your feedback and welcome further discussion.

Best regards, \
The Authors

---

### Decision · Action_Editor_L785 · 2025-11-02

**Recommendation:** Accept as is

**Audience:**

Yes

**Audience Explanation:**

The findings are of particular interest to researchers in reinforcement learning (RL), imitation learning, and robotics. Specifically, the paper appeals to the active online imitation learning community and those working on robotics and autonomous systems due to RIZE's focus on improving robustness and efficiency. Furthermore, the general insight regarding the usefulness of incorporating various sources of aleatoric uncertainty (as achieved through distributional critics) to improve decision-making and optimization is relevant to the broader TMLR audience.

**Claims And Evidence:**

Yes

**Claims Explanation:**

The paper proposes RIZE, a novel Maximum Entropy Inverse Reinforcement Learning (IRL) method that uses adaptive targets for squared TD regularization to impose dynamic bounds on recovered rewards. It integrates distributional RL via an Implicit Quantile Network (IQN) critic and achieved expert-level performance, uniquely solving the challenging Humanoid-v2 task.

Reviewers sought clarification on the theoretical motivation and potential divergence risks of the adaptive regularization scheme, and requested ablations and expansion of the empirical validation. The authors responded by expanding the empirical evaluation to include new tasks (Hopper-v2, Hammer-v1) and competitive baselines (CSIL, BC) using conventional mean returns with 95% confidence intervals. They conducted crucial ablations, including testing HL-Gauss on LSIQ and comparing regularization strategies, and added figures demonstrating that adaptive targets successfully bound implicit rewards, resolving stability concerns.

The final recommendations were positive. Reviewers agreed the claims were supported by clear, correct, and compelling evidence, noting that the revised paper successfully addresses fundamental challenges of high interest to the community.